# Spatio-temporal dynamics of snow cover based on multi-source remote sensing data in China

Xiaodong Huang[1], Jie Deng[1], Xiaofang Ma[1], Yunlong Wang[1], Qisheng Feng[1], Xiaohua Hao[2], Tiangang Liang[1]

[1]Key Laboratory of Grassland Agro-Ecology System, College of Pastoral Agriculture Science and Technology, Lanzhou University, Lanzhou 730020, China;

[2]Chinese Academy of Sciences, Cold and Arid Regions Environmental and Engineering Research Institute, Lanzhou 730000, China

*Correspondence to*: Xiaodong Huang (huangxd@lzu.edu.cn)

**Abstract.** By combining optical remote sensing snow cover products with passive microwave remote sensing snow depth (SD) data, we produced a MODIS cloudless binary snow cover product and a 500 m snow depth product. The temporal and spatial variations of snow cover from December 2000 to November 2014 in China were analyzed. The results indicate that, over the past 14 years, (1) the mean snow-covered area (SCA) in China was 11.3% annually and 27% in winter season, with the mean SCA decreasing in summer and winter seasons, in increasing in spring and fall seasons, and no much change annually; (2) the snow-covered days (SCDs) showed increasing in winter, spring, and fall, and annually, whereas decreasing in summer; (3) the average SD decreased in winter, summer, and fall, while increased in spring and annually; (4) the spatial distributions of SD and SCD were highly correlated seasonally and annually; and (5) the regional differences in the variation of snow cover in China were significant. Overall, the SCD and SD increased significantly in South and Northeast China, decreased significantly in northern Xinjiang Province. The SCD and SD increased on the southwest edge and in the southeast part of the Tibetan Plateau, whereas it decreased in the north and northwest regions.

## 1 Introduction

Snow cover is closely related to human lives, and it has both positive and negative effects (Liang et al., 2004). High and mid-latitude regions contain abundant snow cover and glacial resources, which are the source regions for many rivers (Zhang et al., 2002). Snowmelt runoff can make up more than 50% of the total discharge of many drainage basins (Seidel and Martinec, 2004). Snow cover is an important resource for industrial, agricultural, and domestic water use. Especially in arid and semi-arid regions, the development of agricultural irrigation and animal husbandry relies on the melting of snow cover (Pulliainen, 2006; Li, 2001). Winter water deficiencies can easily cause droughts (Cezar Kongoli et al., 2012). On the other hand, flood disasters caused by melting snow cover and snow disasters such as avalanches, glacial landslides, and snowdrifts are also common (Gao et al., 2008; Liu et al., 2011; Shen et al., 2013).

Rising temperatures due to global warming rapidly change the snow cover conditions in seasonal snow-covered regions, which has led to accelerated melting of most ice sheets and permanent snow covers (Yao et al., 2012), increasing snowline elevations (Chen, 2014), decreasing wetland areas, and

the reallocation of precipitation, which has further led to frequent floods and snow disasters (Lee et al., 2013; Wang et al., 2013). Global warming is an indisputable fact, and rising temperature will strongly affect alpine and polar snow cover (IPCC, 2013). The variation of global and regional snow covers greatly affects the use of snow resources by humanity, and the feedback mechanism of albedo further affects climate (Bloch, 1964; Robinson, 1997; Nolin and Stoeve, 1997). Several studies have indicated that the snow cover in the alpine regions in China affect the atmospheric circulation and weather systems in East Asia and further affect the climate in China (Qian et al., 2003; Zhao et al., 2007). Alpine snow cover has important implications for hydrology, climate, and the ecological environment (Chen and Liu, 2000; Hahn and Shula, 1976).

China is large, and its snow-covered regions are widely distributed geographically. North Xinjiang, Northeast China-Inner Mongolia, and the Tibetan Plateau are the 3 major regions with seasonal snow cover in China (Wang et al., 2009). They are also the major pasturing regions. Winter and spring snowfalls are the major water resources in north Xinjiang and the Tibetan Plateau (Pei et al., 2008; Chen et al., 1991; Wang et al., 2014). Heavy snowfall can also cause severe snow disasters and large numbers of livestock deaths (Liu et al., 2008; Chen et al., 1996). Floods caused by melting snow cover also frequently occur in the spring, severely limit the development of grassland animal husbandry and affect the safety of human lives (Shen et al., 2013). Therefore, accurate acquisition of snow-covered area (SCA) and SD information is significant for understanding climate change and the hydrological cycle, conducting water resource surveys, and preventing and forecasting snow disasters in China.

Recent studies of the distribution and variation of snow cover in China have progressed greatly, but they have mainly focused on the Tibetan Plateau, Xinjiang, and Northeast China (Chen and Li, 2011). Furthermore, the results from different snow cover datasets are slightly different, and the snow cover variations in different regions are also different. MODIS data, which have high spatial and temporal resolution, have been widely used in the remote sensing fields of ecology, atmospheric science, and hydrology. However, clouds strongly interfere with optical sensors. Hence, we cannot directly use snow cover products acquired by optical sensors to effectively quantify SCA. Passive microwaves can penetrate clouds and are not affected by weather. However, the coarse resolution of passive microwave products greatly limits the accuracy of regional snow cover monitoring. Therefore, cloud removal and downscaling are effective approaches for enhancing the accuracy of snow cover monitoring using optical and passive microwave products, respectively.

This study used the MODIS daily snow cover product and passive microwave SD data to produce a daily cloudless SCA product and a downscaled SD product with a 500 m spatial resolution. The integrated daily snow products were used to analyze the temporal and spatial variations of the snow cover in China from December 2000 to November 2014 and quantitatively evaluate the variation of SCA, snow-covered days (SCDs), and average SD to provide a basis for further understanding the interaction between climate and snow cover under the background of globe warming in China.

## 2 Materials and Methods

### 2.1 Study area

China has a large area and a large population, with mountains, plateaus, and hills accounting for ~67%

of the land area, and basins and plains for ~33% (Figure 1). The mountains are mostly oriented east-west and northeast-southwest, including the Altun Mountains, Tianshan, Kunlun, Karakoram, Himalaya, Yinshan, Qinling, Nanling, Daxing'anling, Changbaishan, Taihang, Wuyi, Taiwan, and Hengduan. The Tibetan Plateau, which has an average elevation of more than 4000 m, is located to the southwest and is known as the "Roof of the World". Mount Everest is 8844.43 m in height and is the highest mountain in the world. To the north and east, Inner Mongolia, the Xinjiang area, the Loess Plateau, the Sichuan Basin, and the Yunnan-Guizhou Plateau are second-stage terrains of China. The region from east of the Daxing'anling-Taihang-Wushan-Wuling-Xuefeng Mountains to the shoreline mostly contains third-stage terrains composed of plains and hills with an average elevation of less than 1000 m. The multi-year stable snow cover is mainly distributed in the Tibetan Plateau, Northeast China and Inner Mongolia, and northern Xinjiang covering a total area of ~4,200,000 km$^2$. This snow cover forms the major freshwater reservoirs for most part of China (Li et al., 1983).

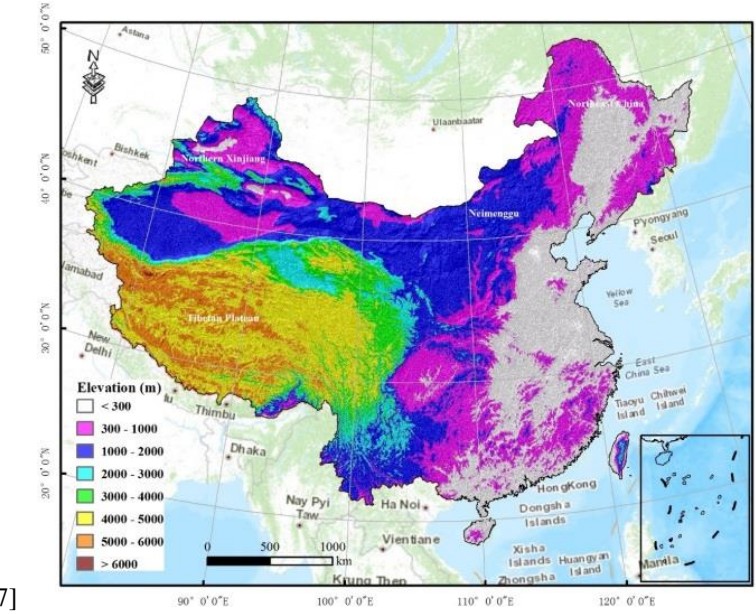

[37]

Figure 1: Schematic diagram of the study region.

## 2.2 Remote sensing snow products

The SD data used in this study were from the 'Environmental and Ecological Science Data Center for West China' (http://westdc.westgis.ac.cn), which is a database with a long time series of SD in China (1979–2014) (Che et al., 2008; 2016; Dai et al., 2012; 2015). It is a daily SD database that was inverted using the brightness and temperature data of the Scanning Multichannel Radiometer (SMMR) (1978–1987), Special Sensor Microwave/Imager (SSM/I) (1987–2007), and Special Sensor Microwave Imager/Sounder (SSMI/S) (2008–2014) passive microwave remote sensing instruments. This product is saved in text format. The unit of SD is cm, and the spatial resolution is 25 km. The database is widely acknowledged and used (Dai et al., 2010; Wang et al., 2013; Bai et al., 2015). The SCA product includes the MOD10A1 and MYD10A1 binary snow cover products of the MODIS/Terra and

MODIS/Aqua daily V005 version covering China (Hall et al. 2002). The data were taken from the National Snow and Ice Data Center (NSIDC). The spatial resolution is 500 m, and the time period is from December 2000 to November 2014.

**2.3 Cloud removal and downscaling algorithms**

Following the MODIS cloud removal algorithm developed by Huang et al., (2014), daily cloudless binary snow cover data were produced for December 2000 to November 2014. The cloud removal algorithm can be summarized in 3 steps: (1) daily snow cover product synthesis: the MOD10A1 and MYD10A1 snow products were combined using the maximum SCA fusion method in accordance with the different acquisition times of the Terra and Aqua satellites and the characteristics of cloud movement; (2) adjacent day analysis: the cloud pixels on a given day were replaced with the pixel values on the previous and following days under cloudless conditions; and (3) combination with the passive microwave SD product: the long time series SD database of China was used to identify cloud pixels, completely reclassify the residual cloud pixels to land or snow pixels, and produce the MODIS daily cloudless binary snow cover images. Based on the downscaling algorithm for the AMSR-E snow water equivalent product by Mhawej et al. (2014), we applied a downscaling algorithm to the passive microwave SD product and built the 500 m spatial resolution SD data of China for December 2000 to November 2014. The equation is as follows:

$$\begin{cases} \text{if MODIS} = 0 \\ SD_{sp} = 0 \\ \text{else} \\ SD_{sp} = \dfrac{SD \times SDY_i \times 2500}{SDT_i}, \end{cases} \qquad (1)$$

where $SD_{sp}$ is the sub-pixel daily SD with a 500 m spatial resolution, $SD$ is the daily SD with a 25 km spatial resolution, $SDY_i$ is the average number of SCDs for each MODIS pixel in year $i$, and $SDT_i$ is the sum of the total SCDs for each SD pixel in year i.

**2.4 Analysis of the snow cover variation**

The Mann-Kendall (M-K) method is a nonparametric test method widely used in the analysis of long time series of data (Helsel and Hirsch, 1992). This method monitors the variation of monotonic nonlinear data. It has no requirement for the data distribution, and it can avoid the interference of a few anomalies (Mcbean and Motiee, 2008). This study used the M-K method to analyze the trend and

significance level of the SCDs and SD in China at the pixel scale. For a series $X_i = (X_1, X_2, \ldots, X_n)$

with n samples, the test process is as follows:

$$Z = \frac{S}{\sqrt{VAR(S)}} \tag{2}$$

where:

$$S = \sum_{i=1}^{n} \sum_{j=i+1}^{n} sgn(X_j - X_i) \tag{3}$$

$$sgn(X_j - X_i) = \begin{cases} +1, if (X_j - X_i) > 0 \\ 0, if (X_j - X_i) = 0 \\ -1, if (X_j - X_i) < 0 \end{cases} \tag{4}$$

$$VAR(S) = \frac{n(n-1)(2n+5) - \sum_{i=1}^{m} t_i(t_i-1)(2t_i+5)}{18} \tag{5}$$

where n is the year count (n = 14), m is the number of nodes (repetitive data groups) in the series, and $t_i$

is the node width (the number of repetitive data points in the i[th] repetitive data group).

When n ≤ 10, we directly used the statistic S for the two-sided trend test. S > 0 represents an increase, S

= 0 represents no variation, and S < 0 represents a decrease. At a given significance level α, if

$|S| \geq S_{\alpha/2}$, the trend of the series is significant; otherwise, it is insignificant.

When n > 10, the statistic S approaches the standardized normal distribution. We used the test statistic

Z for the two-sided trend test. Z > 0 represents an increase, Z = 0 represents no variation, and Z < 0

represents a decrease. At a given significance level α, we looked up the critical $Z_{\alpha/2}$ in the normal

distribution table. If $|Z| > Z_{\alpha/2}$, the series trend is significant; if $|Z| \leq Z_{\alpha/2}$, the trend is insignificant.

Sen's median method was also used to analyze the slope of the variation in the annual SCDs. This

method calculates the median slope of n(n-1)/2 pairs of combinations in a series of length n. The

equation is:

$$\beta = Median\left(\frac{x_i - x_j}{i - j}\right), i > j \tag{6}$$

where β > 0 represents an increase in the trend, and β < 0 represents a decrease in the trend.

**3 Results**

**3.1 Snow-Covered Area**

Fig. 2 summarizes the average annual SCA between 2001 and 2014. Leap years occurred in 2004, 2008

and 2012, so the average SCA refers to the mean of 366 days for these years. The results indicated that

the average annual SCA in China in 2001–2014 constituted 11.3% of the entire study region. The

average annual SCA varied slightly over the past 14 years, but did not increase or decrease

significantly.

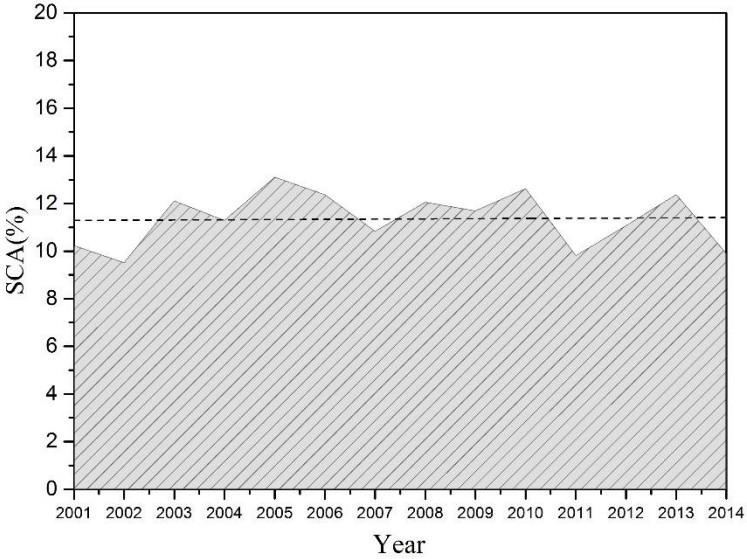

Figure 2: Average annual SCA in China between 2001 and 2014.

Fig. 3 summarizes the average SCA during each season in China from December 2000 to
November 2014. The results indicated that over the past 14 years, the average SCA in China was
approximately 27.0% during the winter, 10.7% during the spring, 6.8% during the fall, and 1.2% during
the summer. The average SCA during the winter and summer decreased and the average SCA in spring
and fall increased.

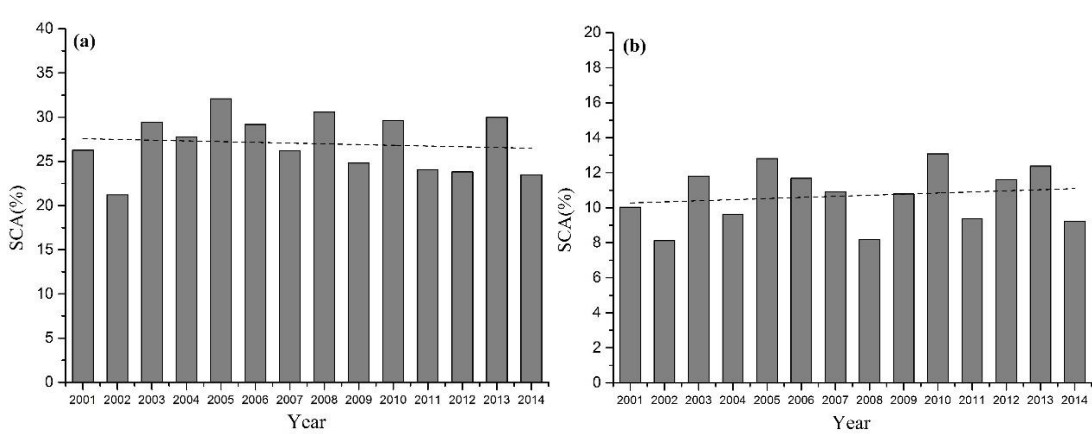

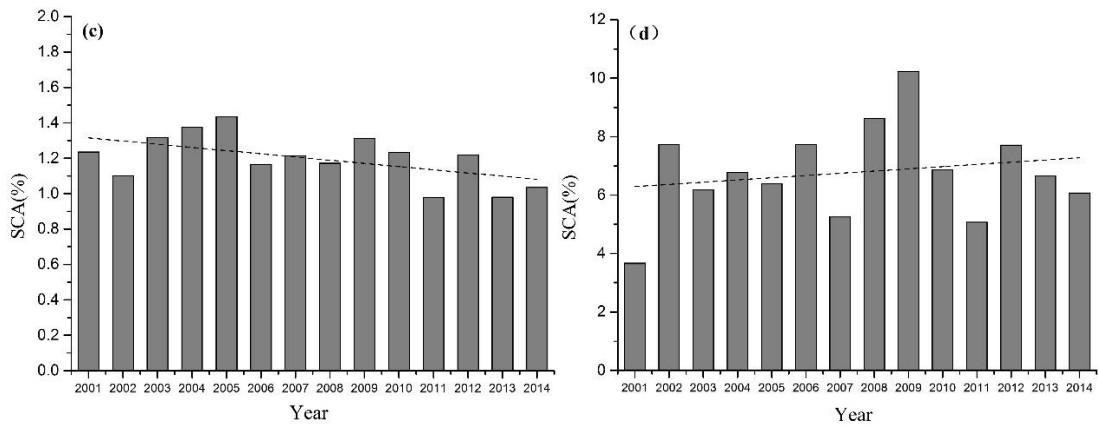

Figure 3: Histograms of the average SCA in each season in China from December 2000 to November 2014. (a), (b), (c), and (d) are the average SCA in winter, spring, summer, and fall, respectively.

**3.2 Snow-Covered Days**

Fig. 4 shows the spatial distribution of the average annual number of SCDs from December 2000 to November 2014. The transient snow-covered regions with less than 10 annual SCDs were distributed primarily in East and South China, the Tarim Basin in Xinjiang, the Badian Jaran Desert in Inner Mongolia, and the Qaidam Basin in the Tibetan Plateau. The unstable snow-covered regions ($10 < SCD \leq 60$) were primarily located to the north of the Hengduan, Qinling - Taihang and Changbai Mountains in China, in the North China Plain, in some hilly areas in Southeast China, and most regions in the north and west parts of China. The relatively stable snow-covered regions ($60 < SCD \leq 350$) were primarily located in Northeast China-Inner Mongolia, north Xinjiang, and the high mountains of the Tibetan Plateau. Considering the accuracy of the MODIS snow product (Wang et al., 2015), we classified regions with more than 350 SCDs as permanent snow-covered regions; they were mainly located in the Tienshan Mountains in Xinjiang, the Qilian, Kunlun, Nyainqentanglha, and the Himalaya in the Tibetan Plateau.

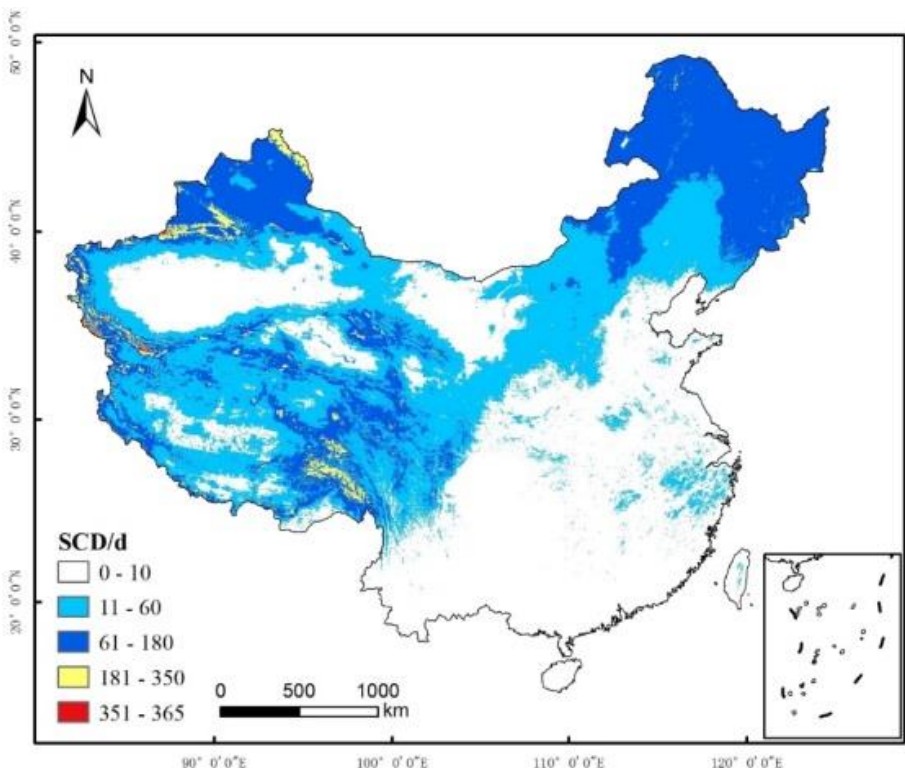

Figure 4: Spatial distribution of the average annual number of snow-covered days during 2001-2014 in China.

The M-K method was used to analyze the variation in the annual number of SCD in China form December 2000 to November 2014 (Fig. 5). The number of SCDs in China decreased by 29.2% over the past 14 years ($Z < 0$), for which 6.5% of the area decreased significantly ($p < 0.05$). These regions were primarily located in the Tienshan Mountains in Xinjiang and most of the Tibetan Plateau. The regions with increasing numbers of SCDs represented 34.5% of China ($Z > 0$), of which 10.8% of the area increased significantly ($p < 0.05$). These regions were mainly located in the Great Khingan Mountains, Lesser Khinan Mountains, and Changbai Mountains in the northeast part of the country and most regions of South China.

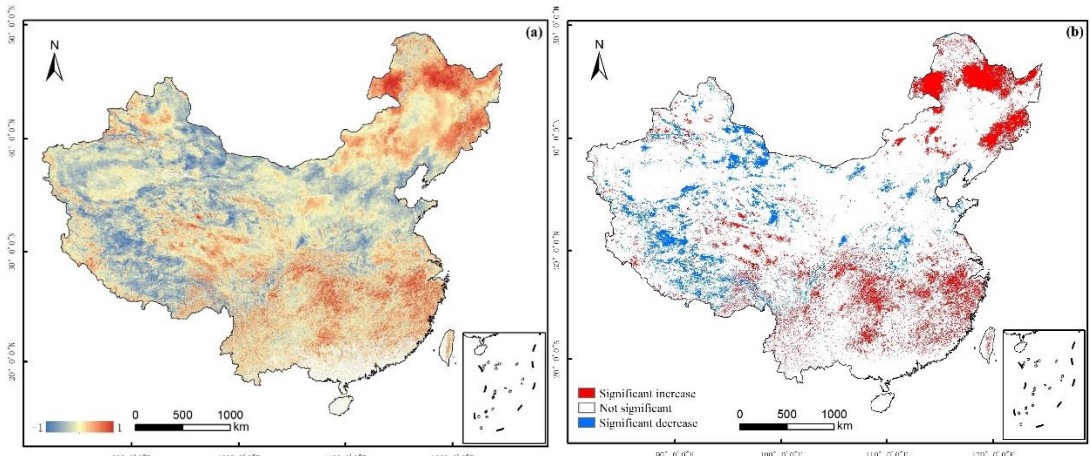

Figure 5: Variation in the average annual SCDs in China based on the Mann-Kendall method from 2001-2014. (a) Variation in the annual SCDs; (b) significance of the variation in the annual SCDs.

The M-K method was also used to analyze the variation in SCDs during the winter (December–February), spring (March–May), summer (June–August), and fall (September–November) in the grid cells (Fig. 6). The results indicated that over the past 14 years, the regions with significantly decreased numbers of winter SCDs represented 5.7% of the area of China, whereas the areas with significant increases made up 7.2% of the study region (Fig. 6(a)). The regions with significantly decreased spring SCDs constituted 4.0% of China, whereas the regions with significant increases represented 6.2% (Fig. 6(b)). The regions with significantly decreased summer SCDs made up 3% of China, whereas the regions with significant increases constituted 2.9% (Fig. 6(c)). The regions with significantly decreased fall SCDs constituted 1.8% of China, whereas the regions with significant increases constituted 5.7% (Fig. 6(d)). The results indicated that over the past 14 years, the summer SCDs decreased, whereas the SCDs during the winter, spring, and fall all increased. The spatial distributions of the increases and decreases in SCDs during each season were highly consistent. Specifically, the winter SCDs in South China increased, the SCDs in Northeast China increased during all of the seasons, and the SCDs in the Xinjiang regions mainly decreased. The SCDs on the southwest margin of the Tibetan Plateau and the southeast region increased, whereas those in the north and northwest mainly decreased.

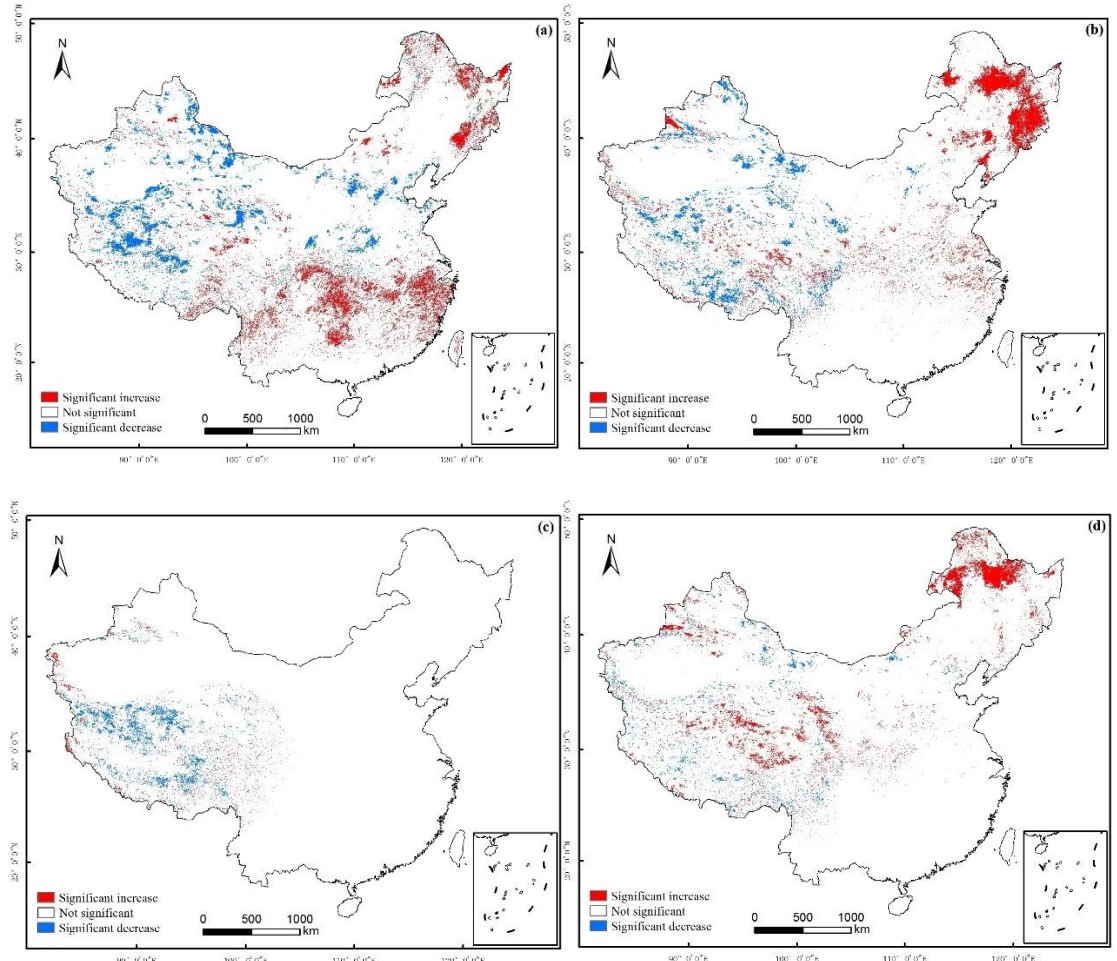

Figure 6: Variation in the number of SCDs during each season in China based on the Mann-Kendall method from 2001to 2014. (a), (b), (c) and (d) show the significance of the variation in the number of

215 SCDs during the winter, spring, summer, and fall, respectively.

The results of the M-K variation analysis showed that the annual number of SCDs in South China increased significantly. To further analyze the trend of the SCDs in China over the past 14 years, we calculated the slope of the variation in the annual SCDs using Sen's median method (Fig. 7). The results indicated that the annual number of SCDs decreased over approximately 22.1% of China ($\beta < 0$),

and increased over 23.5% of China ($\beta > 0$). The rate of decrease in the annual SCDs was less than 2 d/year over 18.5% of the area, which was sparsely distributed in Xinjiang, the Tibetan Plateau, and North China. The rate of decrease in some regions of the Tibetan Plateau exceeded 6 d/year. The rate of increase in the annual SCDs was less than 2 d/year in 18.3% of the area, which was mainly distributed in South China, Northeast China, central northern Xinjiang, and the southeast Tibetan Plateau. The

regions with rates of increase of more than 6 d/year were sparsely distributed in Northeast China and the southeast Tibetan Plateau. The results from Sen's median method were highly consistent with the

results from the Mann-Kendall method, especially in terms of the spatial distribution of the variations. Because the regions in Southeast China were mainly transient snow-covered regions, and the main relatively stable snow-covered regions were located in north Xinjiang, in Northeast China, and the Tibetan Plateau, the variation in the number of SCDs indicated that the annual SCDs in China increased overall (Figs. 5 and 7). However, the decreases in the SCDs in the Tibetan Plateau and the Tienshan Mountains in Xinjiang, which have high elevations, were significant.

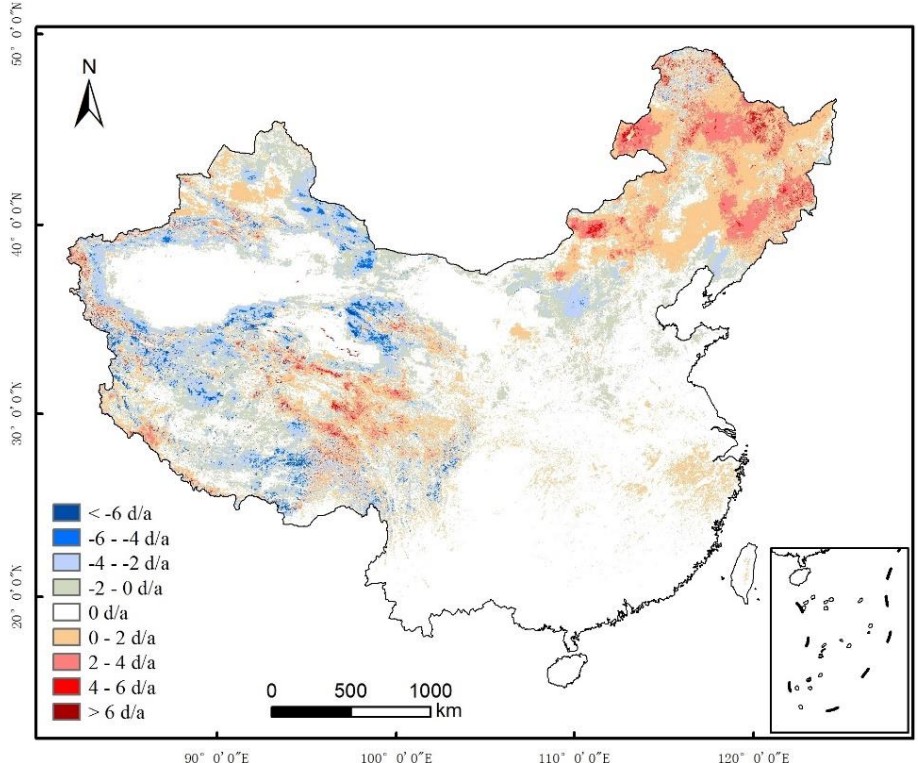

Figure 7: Variation slope of the average annual number of SCD in China based on Sen's median method during the period of 2001-2014.

## 3.3 Snow Depth

SD is a key factor that reflects the variation of the surface snow cover and has important hydrological, climate, and ecological significance. Fig. 8 shows the spatial distribution of the average SD from December 2000 to November 2014. The average SD was calculated by dividing the sum of the SD by the total number of days. The spatial distributions of the average SD and SCDs in China were highly consistent. The regions with high values of the average SD were mainly distributed in the Great Khingan Mountains and Lesser Khinan Mountains in Northeast China, the Altai and Tienshan Mountains in Xinjiang, and the Kunlun and Nyainqentanglha Mountains in the Tibetan Plateau. The multi-year average SD was greater than 7 cm. Except for the Tibetan Plateau, the SD generally

increased with increasing latitude and elevation. However, because of the limited capability of passive microwave data to detect shallow snow and wet snow, the data did not capture any snowfall information in most regions in South China. Therefore, it is necessary to combine optical and passive microwave data to improve the accuracy of snow cover monitoring.

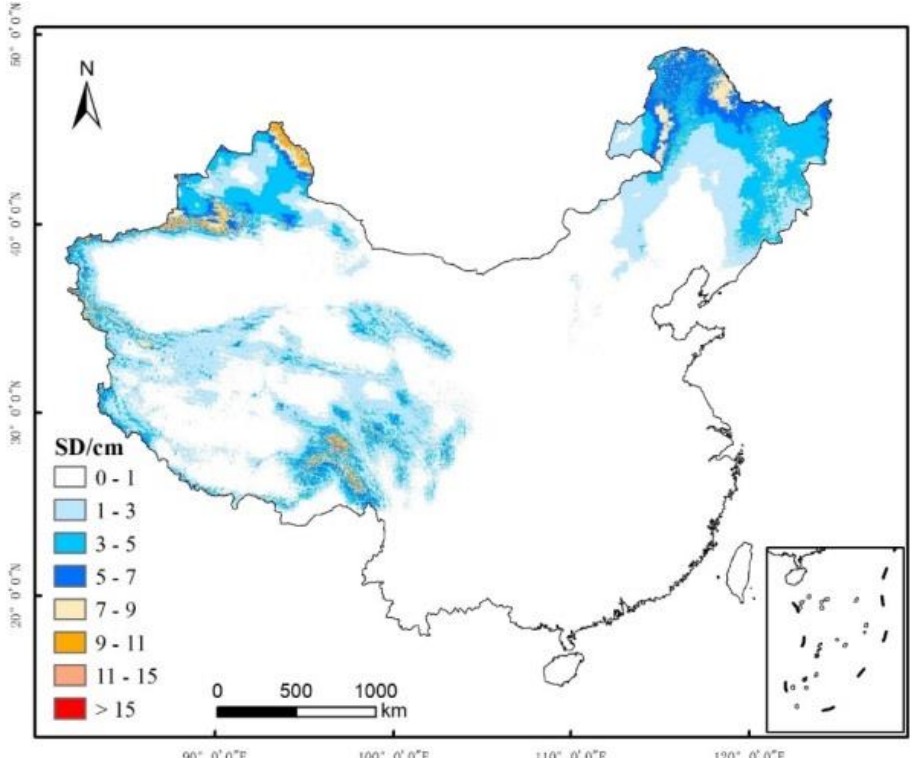

Figure 8: Spatial distribution of the average annual snow depth in China from December 2000 to November 2014.

Fig. 9 summarizes the spatial variation of the average annual SD in China from December 2000 to November 2014. The variations in the SD and the SCDs were highly spatially consistent. The regions with decreasing average annual SD covered 11% of China, and the average annual SD significantly

decreased in 3.3% of the China ($p < 0.05$), primarily in most regions of north Xinjiang and the north Tibetan Plateau. A total of 22.4% of the area showed an increase tendency, and significant increases were observed in 8.5% of the area ($p < 0.05$), which was mainly distributed in Northeast China, the Tienshan and Altai Mountains in Xinjiang, the south Tibetan Plateau, and the Kunlun Mountains.

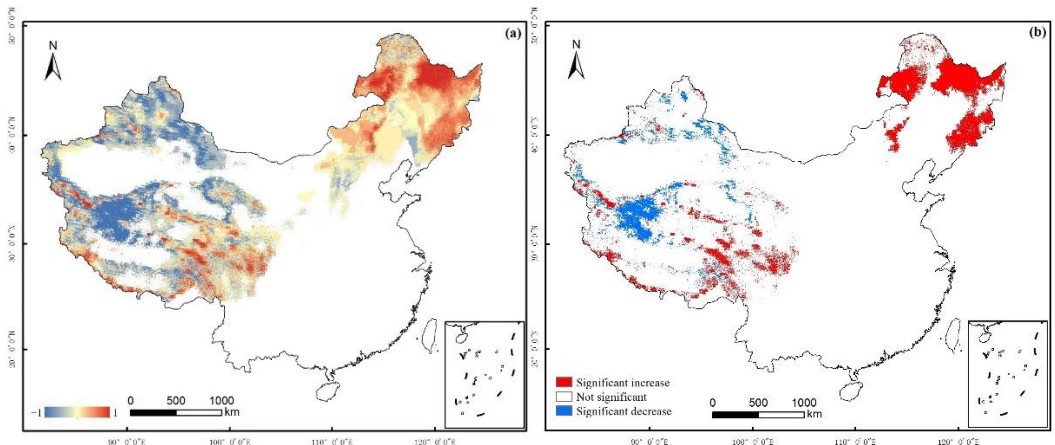

Figure 9: Variation in the average annual SD in China based on the Mann-Kendall method between 2001 and 2014. (a) Variation in the average annual SD; (b) significance of the variation in the average annual SD.

Over the past 14 years, the regions with significantly decreased winter SD made up 10.6% of China, whereas the regions with significant increases constituted 9.3% (Fig. 10(a)). The regions with significantly decreased spring SD constituted 7.9% of the area of China, whereas the regions with significant increases made up 9.8% (Fig. 10(b)). The regions with significantly decreased summer SD made up 1.9% of China, whereas the regions with significant increases constituted 0.9% (Fig. 10(c)). The regions with significantly decreased fall SD represented 7.8% of the area of China, whereas the regions with significant increases only made up 1.8% (Fig. 10(d)). The regions with significantly increased and decreased average SD during the winter and spring were similar. The regions with increased SD were mainly concentrated in Northeast China and the high mountains of the Tibetan Plateau, whereas the regions with decreased SD were primarily located in the hinterlands of Xinjiang and the Tibetan Plateau. The SD during the fall and summer mainly decreased, and the regions with decreasing SD were mainly distributed in Xinjiang and the Tibetan Plateau.

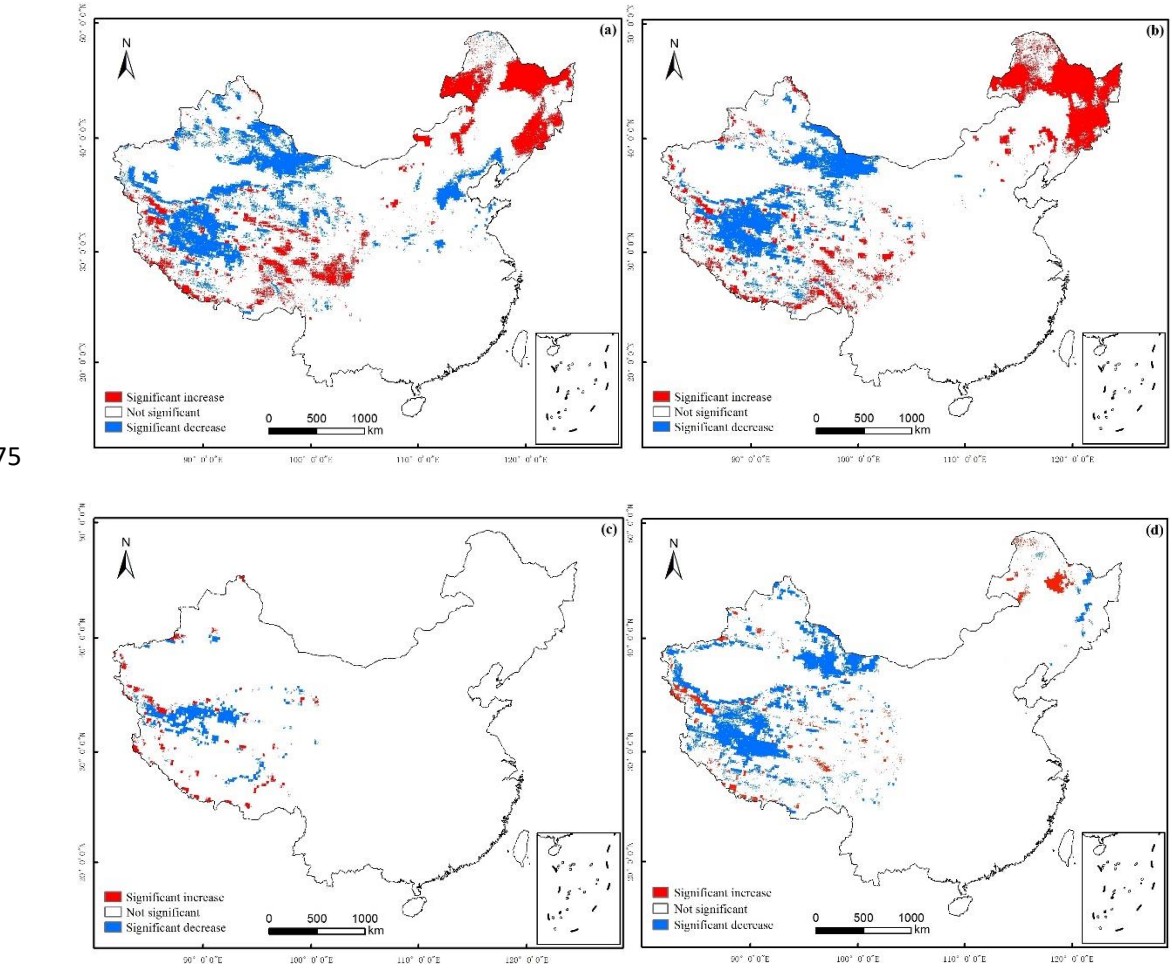

Figure 10: Variation in the average SD during each season in China based on the M-K method from 2001to 2014. (a), (b), (c), and (d) show the significance of the variations during the winter, spring, summer, and fall, respectively.

## 4 Discussion

Snow cover is widely distributed in China. The results of this study indicated that the average annual SCA did not change significantly. The relative stable snow-covered regions (60 < SCD ≤ 350) in China were primarily located in Northeast China-Inner Mongolia, north Xinjiang, and the high mountains in the Tibetan Plateau, and the stable snow area did not change significantly during 2001-2014. Liu et al. (2012) studied the spatial stability of the three major snow-covered regions in China for 2001–2010 and analyzed the characteristics of the seasonal and annual snow cover variations. The results indicated that the snow cover stabilities in the three major snow-covered regions were in the order of Xingjiang > Northeast China-Inner Mongolia > Tibetan Plateau. The stable SCA in China did not change significantly. Same results also found for the relative stable snow-covered regions, whereas the

unstable snow-covered regions (SCDs < 60) had large annual variations in the SCA (Wang et al. 2012).

Dou et al. (2010) used the MODIS snow cover product to study the Tianshan Mountains in China, and indicated that the snow cover in the Tianshan Mountains increased slightly; the increase was especially significant in the winter. Furthermore, the snow cover decreased in the regions at with elevation of ≥ 4000 m and increased in the regions with elevation of < 4000 m. This study found similar results, but

the significant increase in SCDs was observed in the spring, not in the winter.

Dai et al. (2010) indicated that the number of SCDs and the SD in China increased between 1978 and 2005. The western Tibetan Plateau was a sensitive region with an abnormal variation in SCDs, whereas north Xinjiang, the mountainous regions in Northeast China and the east-central Tibetan Plateau were sensitive regions with abnormal SD variations. Che et al. (2005) used the SD data that were inverted

from SSM/I passive microwave data to analyze the snow cover distribution and variations in China for 1993–2002. The results indicated that the snow cover reservoir in China did not increase or decrease significantly over that ten-year period. The winter snow cover reservoir was mainly located in the three major stable snow-covered regions of Xinjiang, the Tibetan Plateau, and Northeast China. The study by Basang et al. (2012) on the variation of snow cover in Tibet indicated that from 1980 to 2009, the SCDs

and maximum SD in Tibet decreased. The decrease was very significant after the start of the 21$^{th}$ century. The variations were slightly different in different seasons, and the results observed by different remote-sensing satellites were also different. Our study showed that over the past 14 years, the SCDs and SD decreased primarily in the hinterlands of the Tibetan Plateau, and increased in the southwest and southeast margins of the Tibetan Plateau. Studies based on long time series of observations by

ground stations have indicated that the number of SCDs and the SD in Northeast China increased every year (Chen and Li, 2011; Yan et al., 2015; Ke et al., 2016), which is consistent with our results for Northeast China over the past 14 years.

**5 Conclusion**

In this study, we used the daily cloudless snow cover and snow depth products both at 500 m pixel size to investigate the variations of the snow-covered area (SCA), snow covered days (SCDs) and snow depth (SD) in China from December 2000 to November 2014. The important results are summarized below:

     (1) The perennial average annual SCA in China was 11.3% over the entire year and 27.0% during

the winter for2001–2014. The average SCA decreased during the winter and summer, and increased

during the spring and fall. The average annual SCA varied slightly, but did not increase or decrease

significantly.

(2) The transient snow cover in China was mainly located in East and Southeast China and some

regions of Xinjiang and Inner Mongolia, whereas the unstable snow-covered regions were distributed

in most of the northern and western regions in China. The stable snow-covered regions were mainly

located in Northeast China-Inner Mongolia, north Xinjiang, and the Tibetan Plateau. The west Tienshan

Mountains in Xinjiang and the mountainous areas of the Tibetan Plateau were the main regions with

permanent snow cover.

(3) The summer SCDs in China decreased, whereas the SCDs increased duirng the winter, spring,

and fall. Specifically, the winter SCDs in South China increased, the SCDs in Northeast China

increased during all of the seasons, and the SCDs in the Xinjiang regions mainly decreased. Overall,

the SCDs increased during 2001-2014 in China.

(4) The spatial distribution of the variation in the average SD was highly consistent with that of

the SCDs. The spatial distributions of the amounts of increase and decrease in the snow cover during

each season were also highly consistent. However, the regional differences in the increases in the

average annual SCDs and average SD were significant. The regions with increasing SCDs and SD were

mainly located in Northeast China, whereas the Tibetan Plateau and Xinjiang were the main regions

with decrease.

Acknowledgements. This work was supported by the China State Key Basic Research Project

(2013CBA01802) and the Chinese Natural Science Foundation Projects (41671330, 31372367, and

41471291).

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

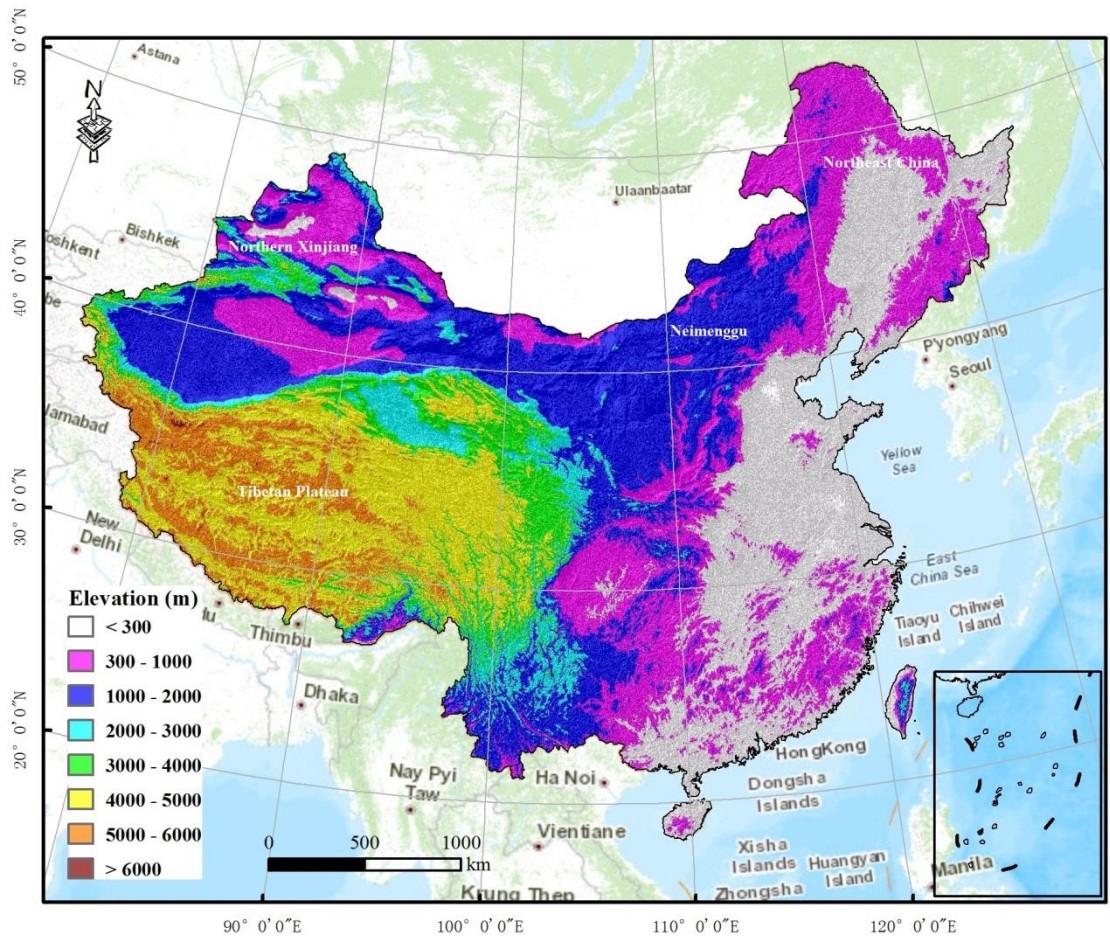

Figure 1: Schematic diagram of the study region.

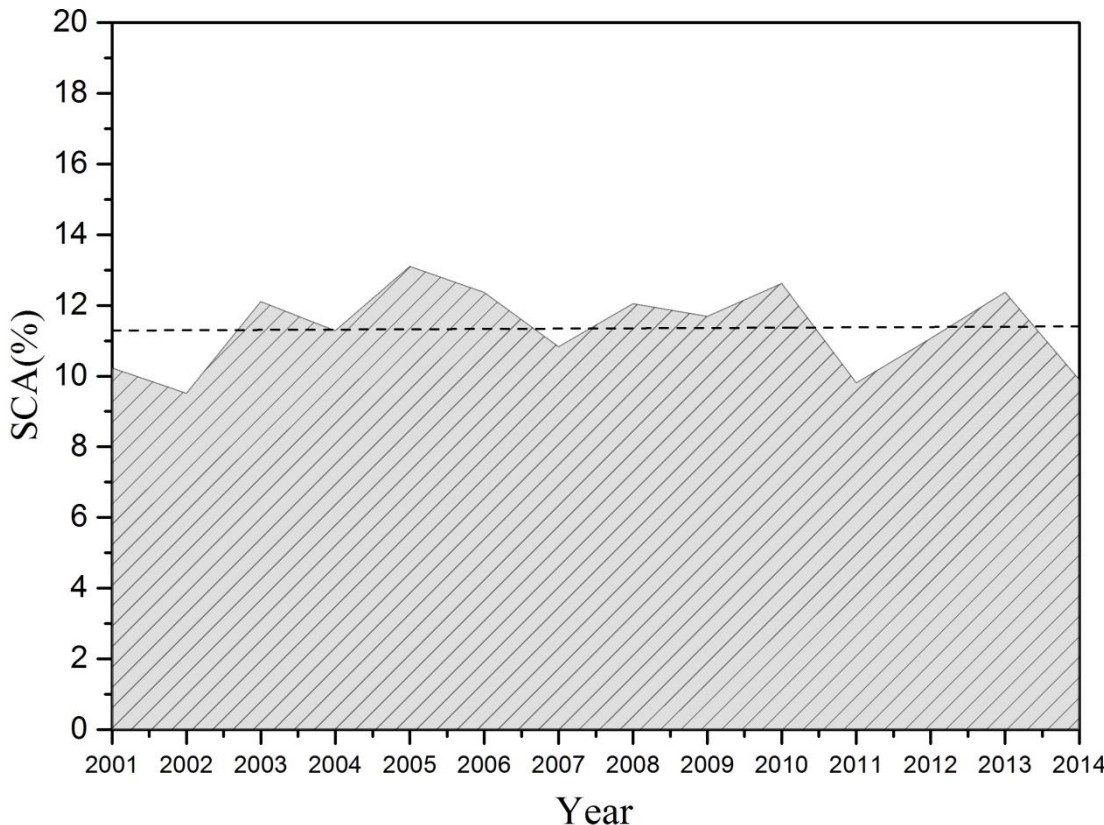

Figure 2: Average annual SCA in China between 2001 and 2014.

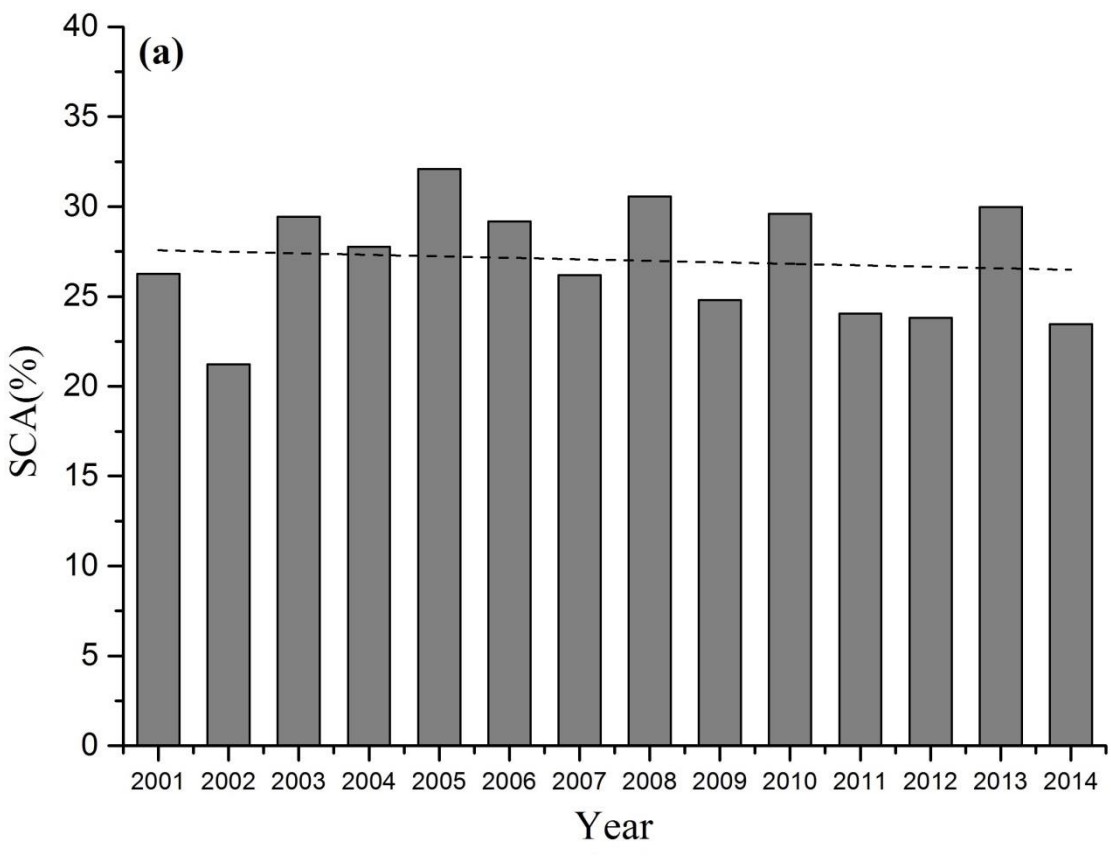

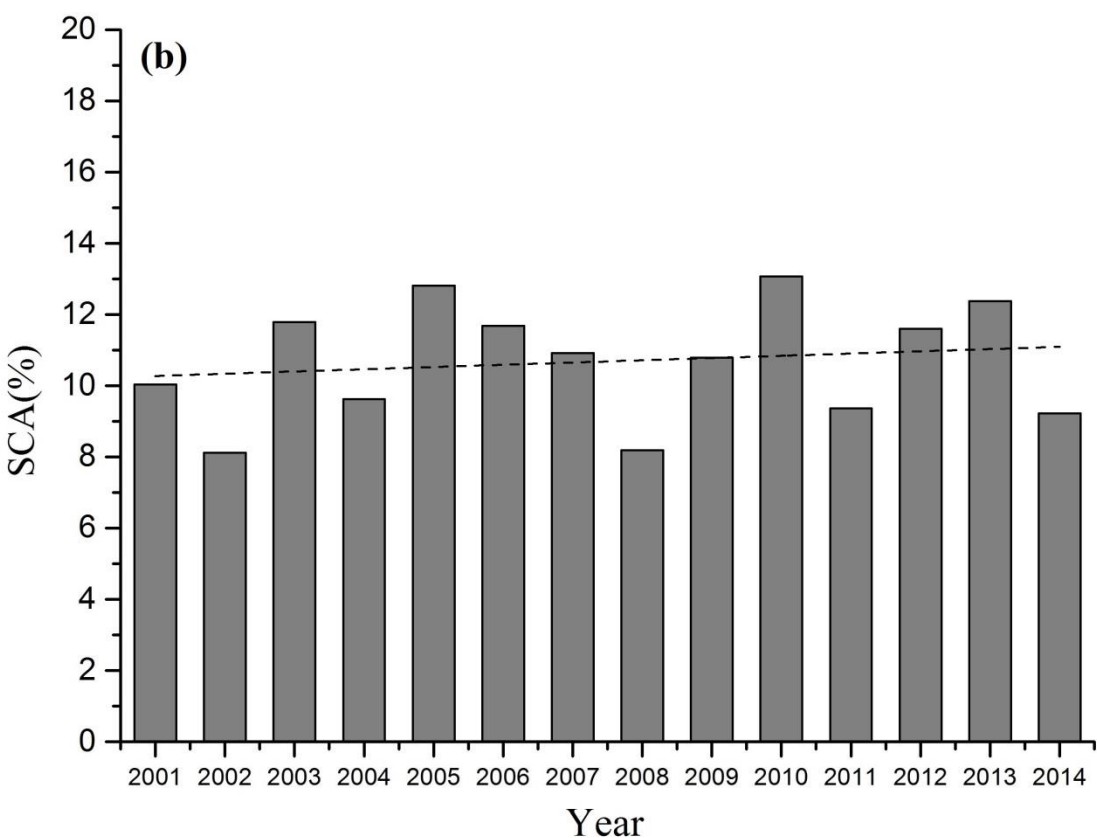

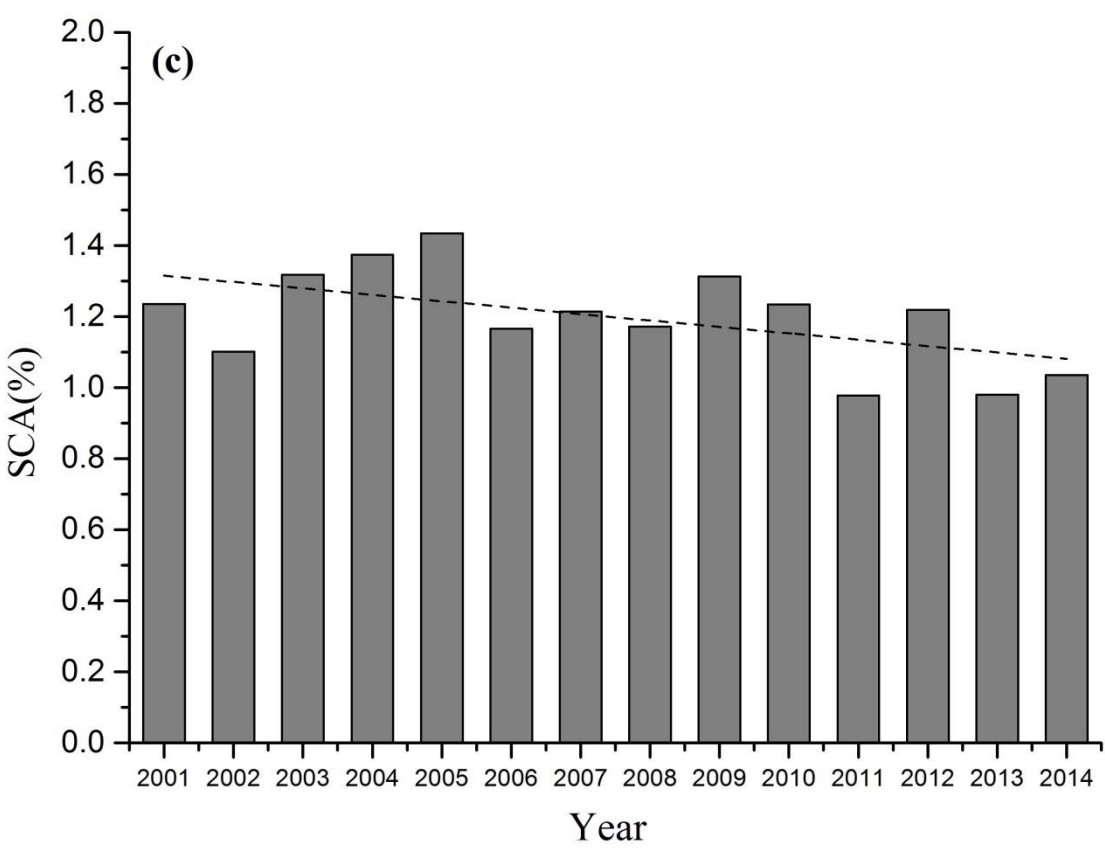

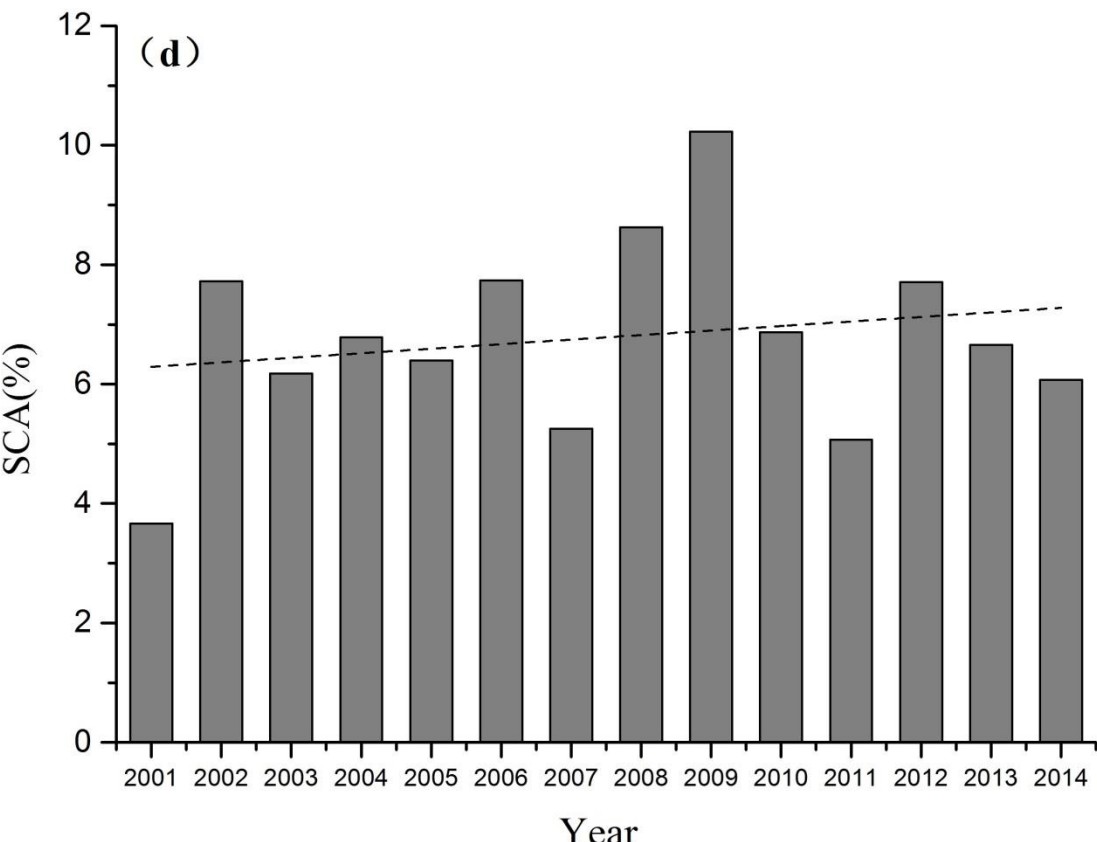

Figure 3: Histograms of the average SCA in each season in China from December 2000 to November
2014. (a), (b), (c), and (d) are the average SCA in winter, spring, summer, and fall, respectively.

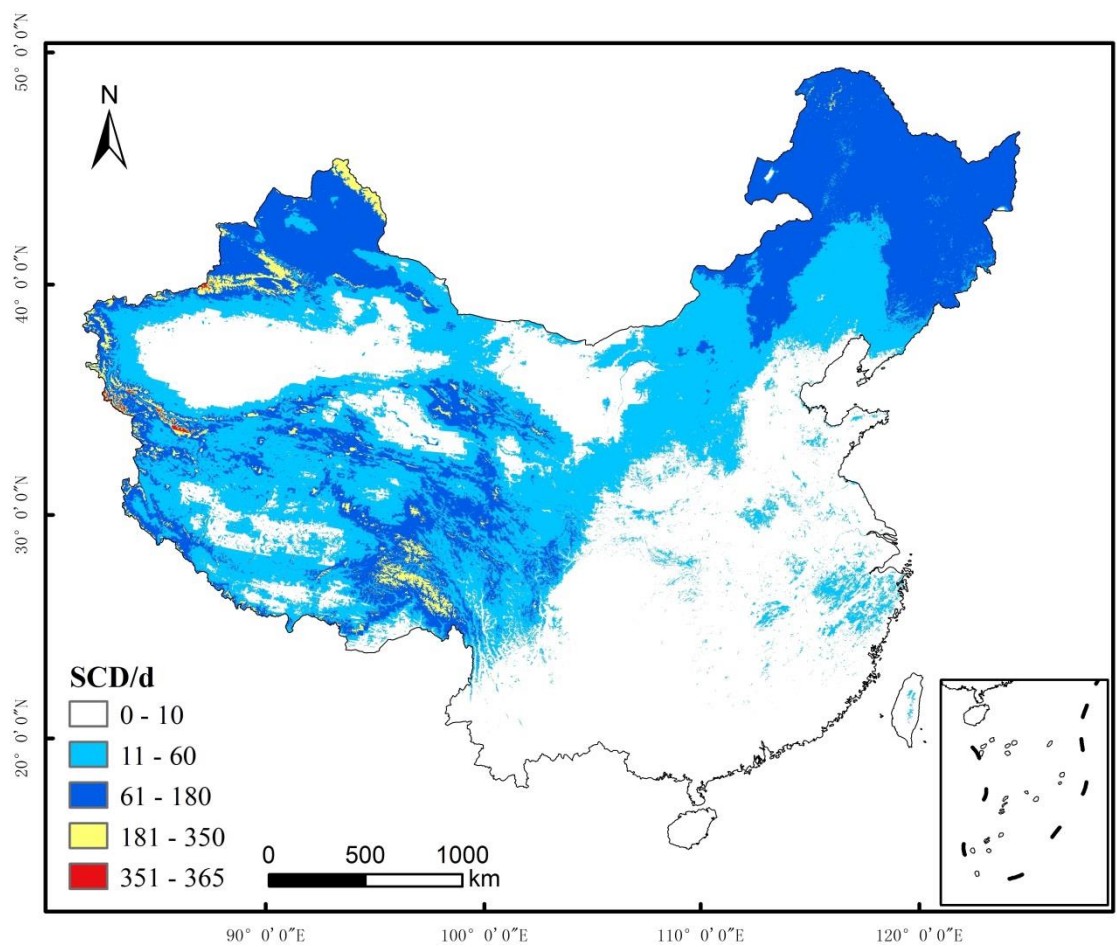

Figure 4: Spatial distribution of the average annual number of snow-covered days during 2001-2014 in China.

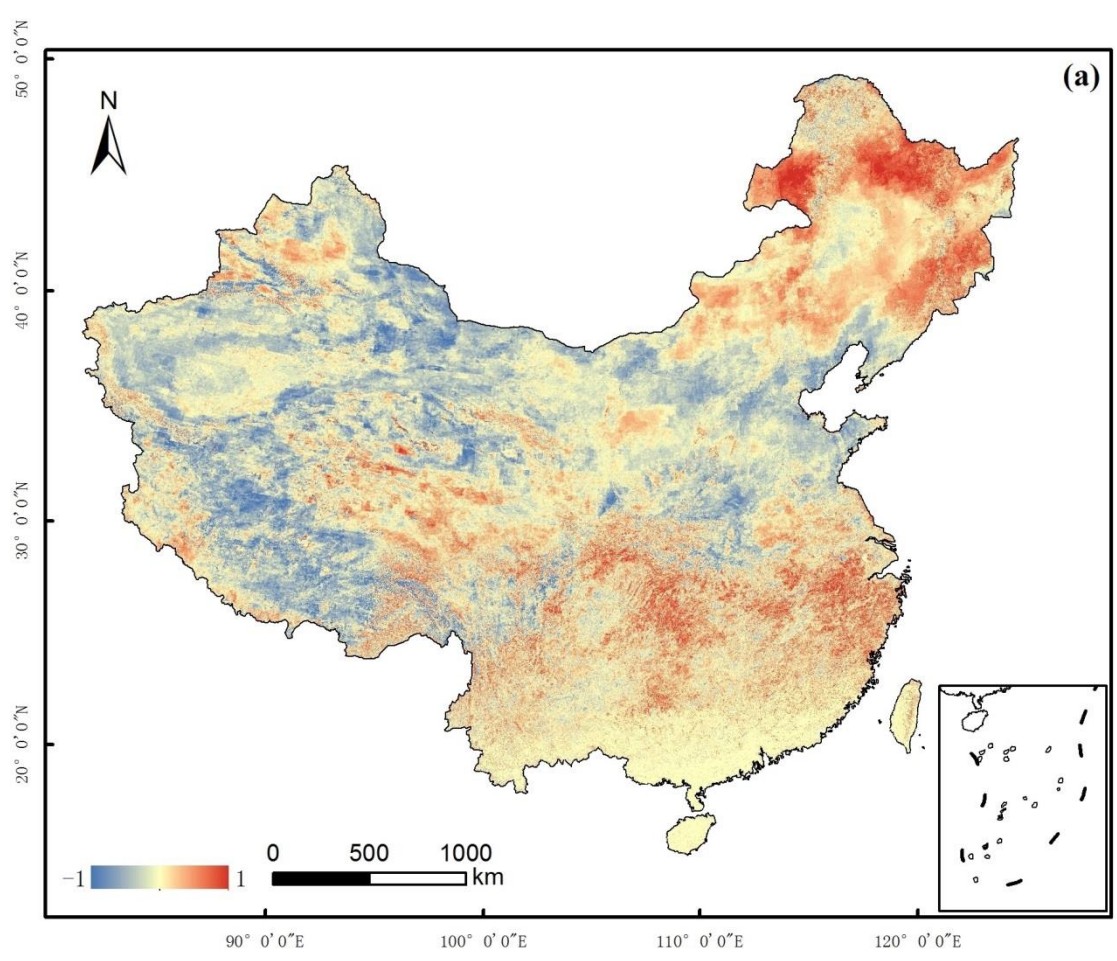

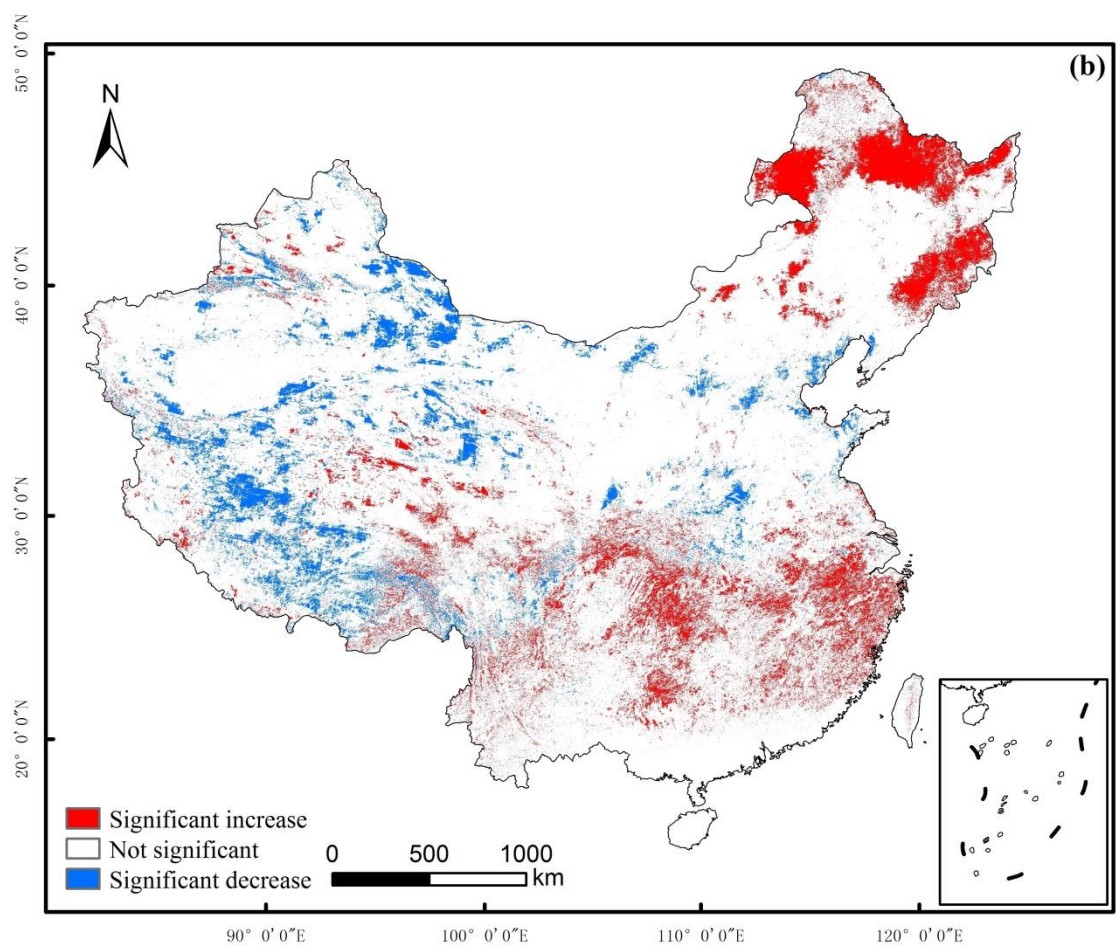

Figure 5: Variation in the average annual SCDs in China based on the Mann-Kendall method from 2001-2014. (a) Variation in the annual SCDs; (b) significance of the variation in the annual SCDs.

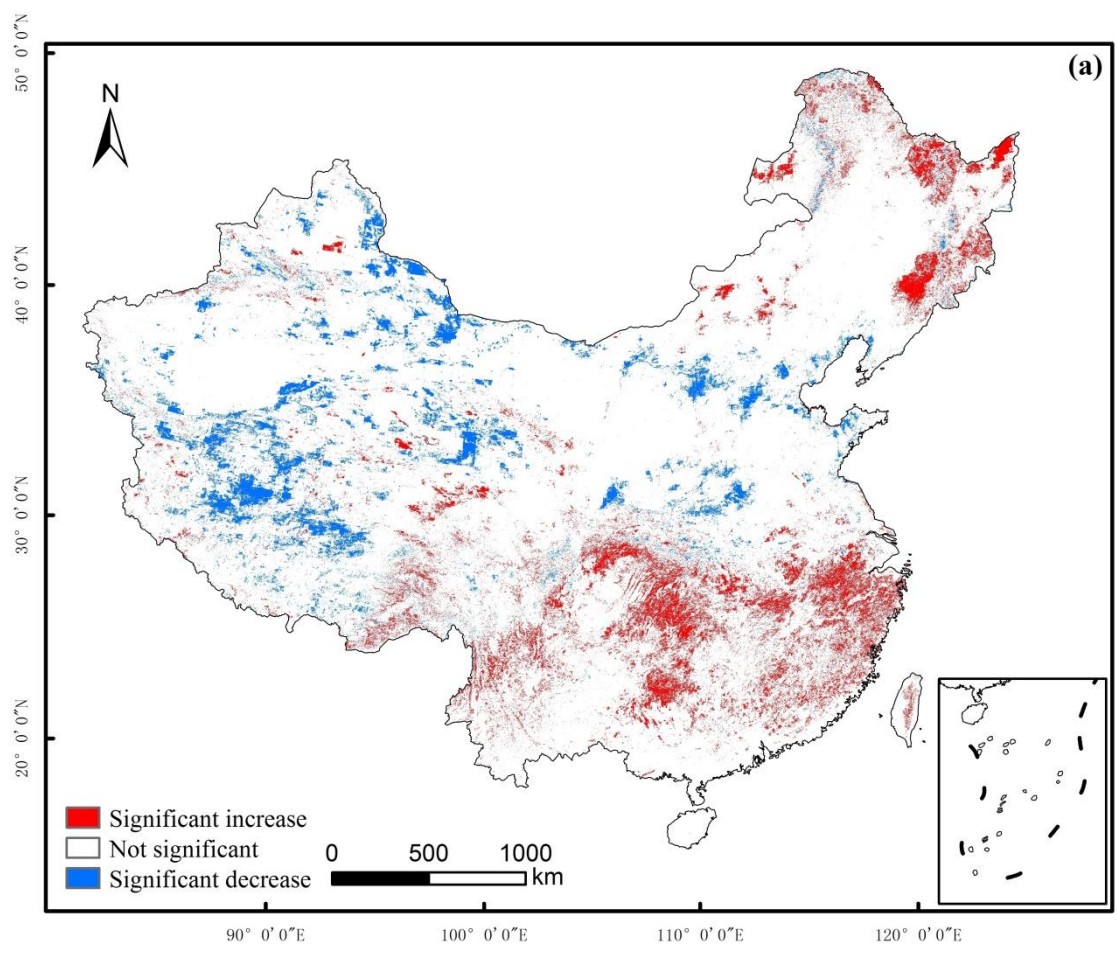

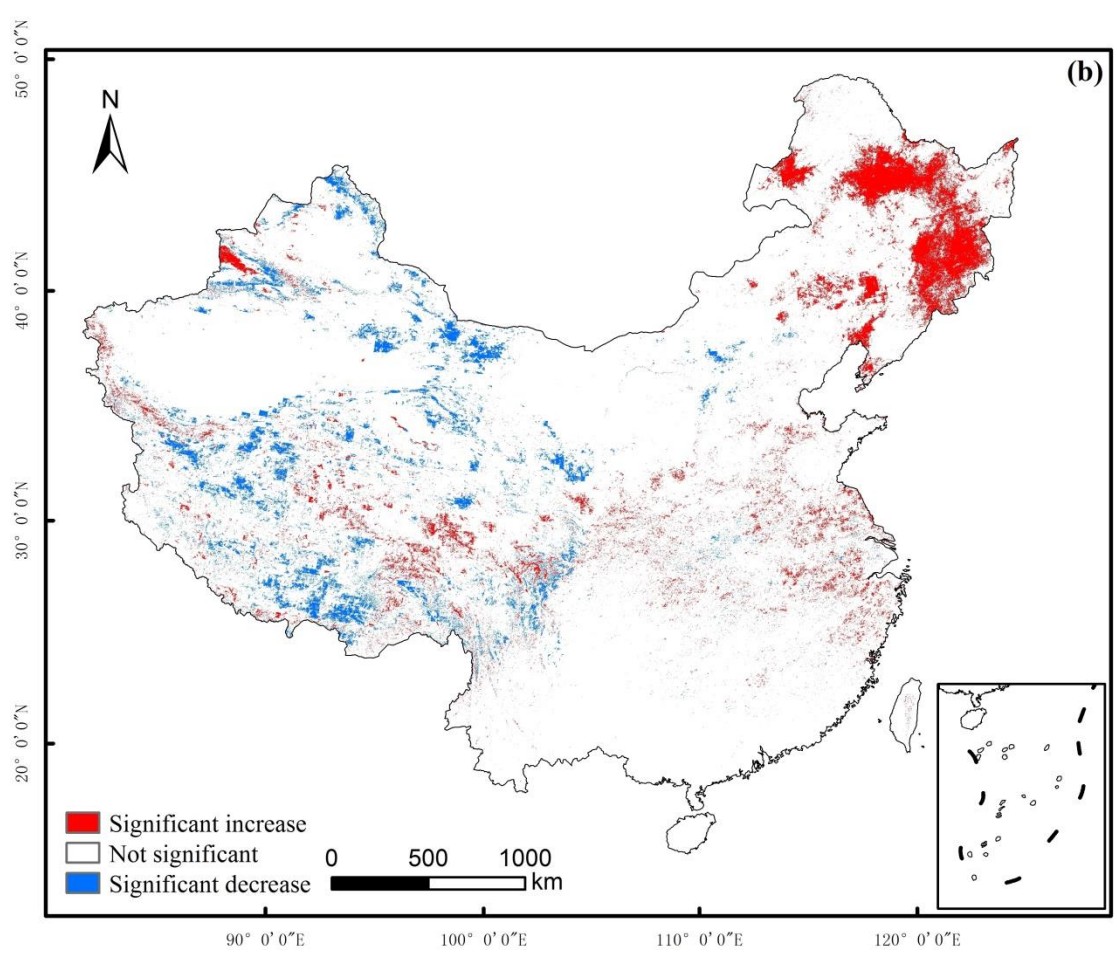

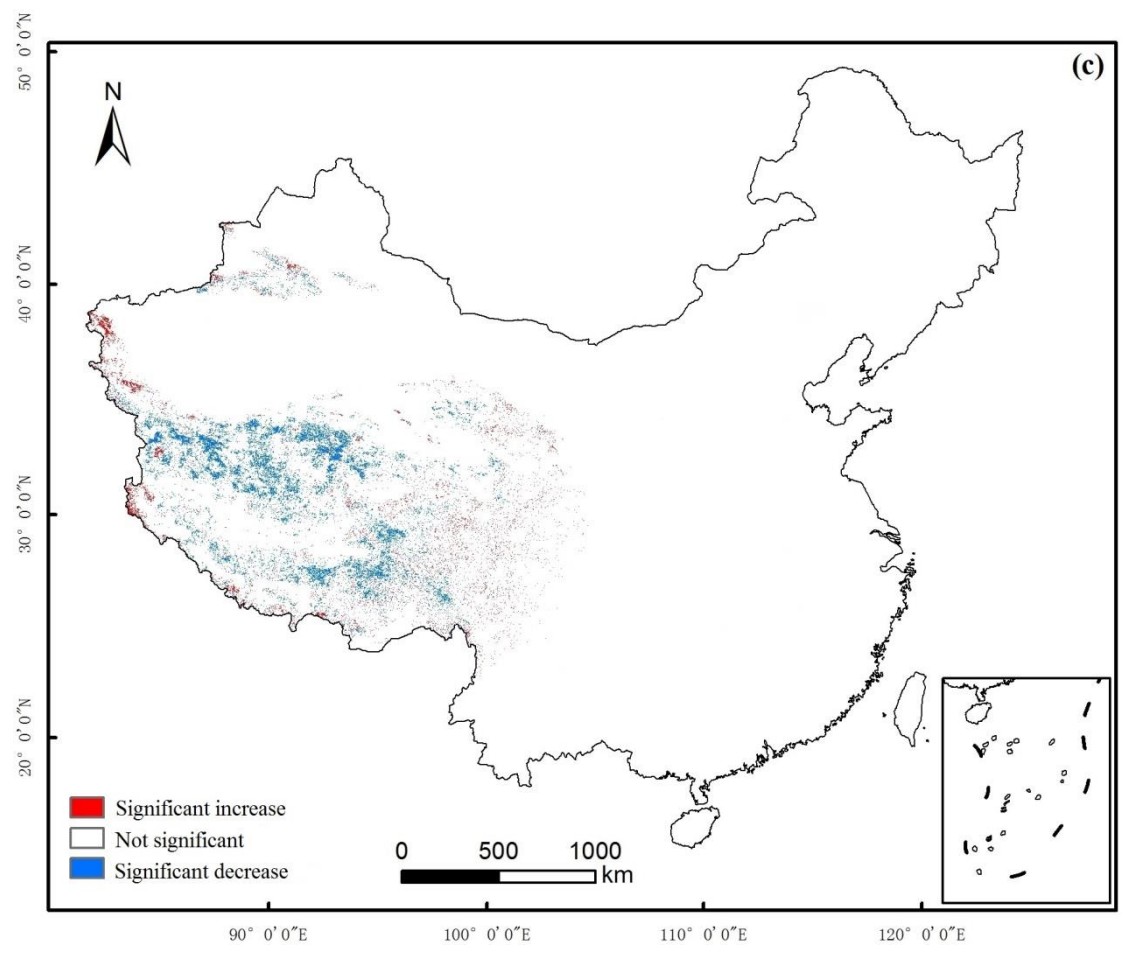

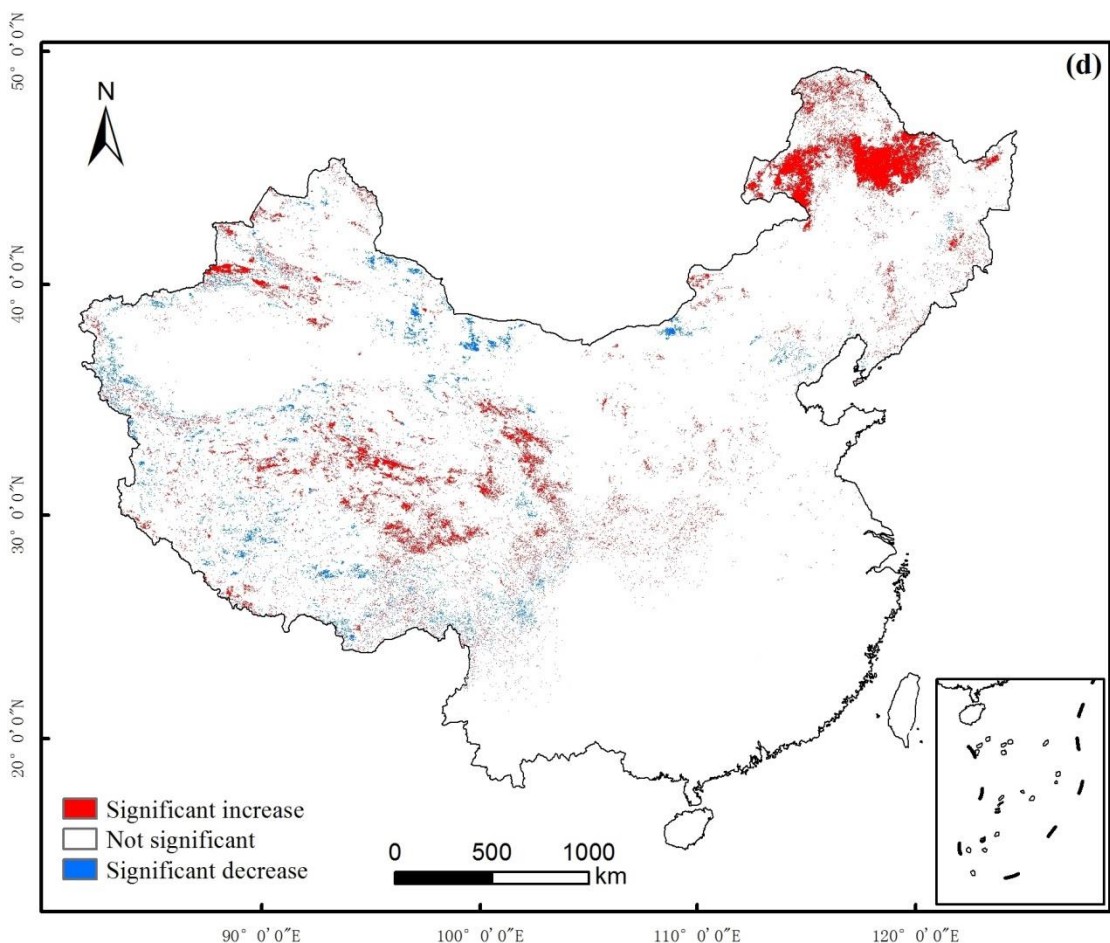

Figure 6: Variation in the number of SCDs during each season in China based on the Mann-Kendall method from 2001to 2014. (a), (b), (c) and (d) show the significance of the variation in the number of SCDs during the winter, spring, summer, and fall, respectively.

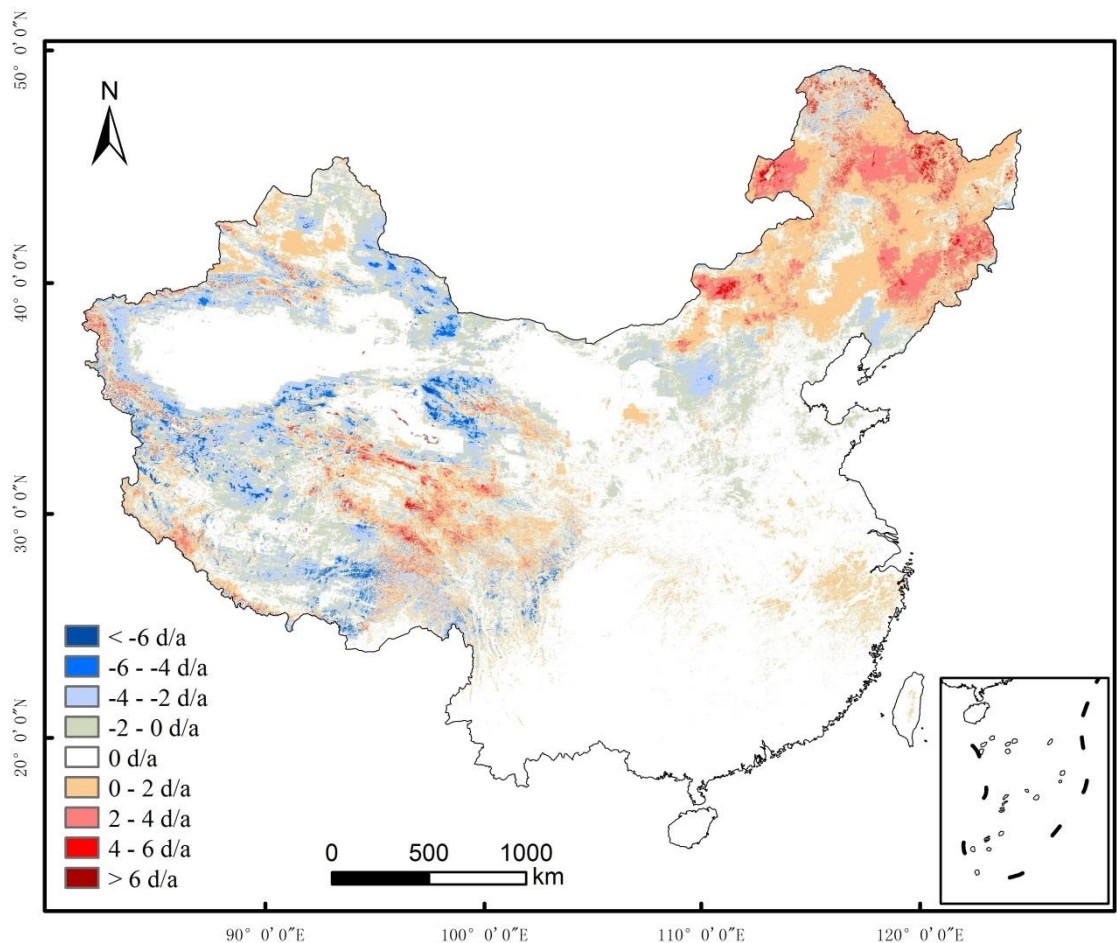

Figure 7: Variation slope of the average annual number of SCD in China based on Sen's median
        method during the period of 2001-2014.

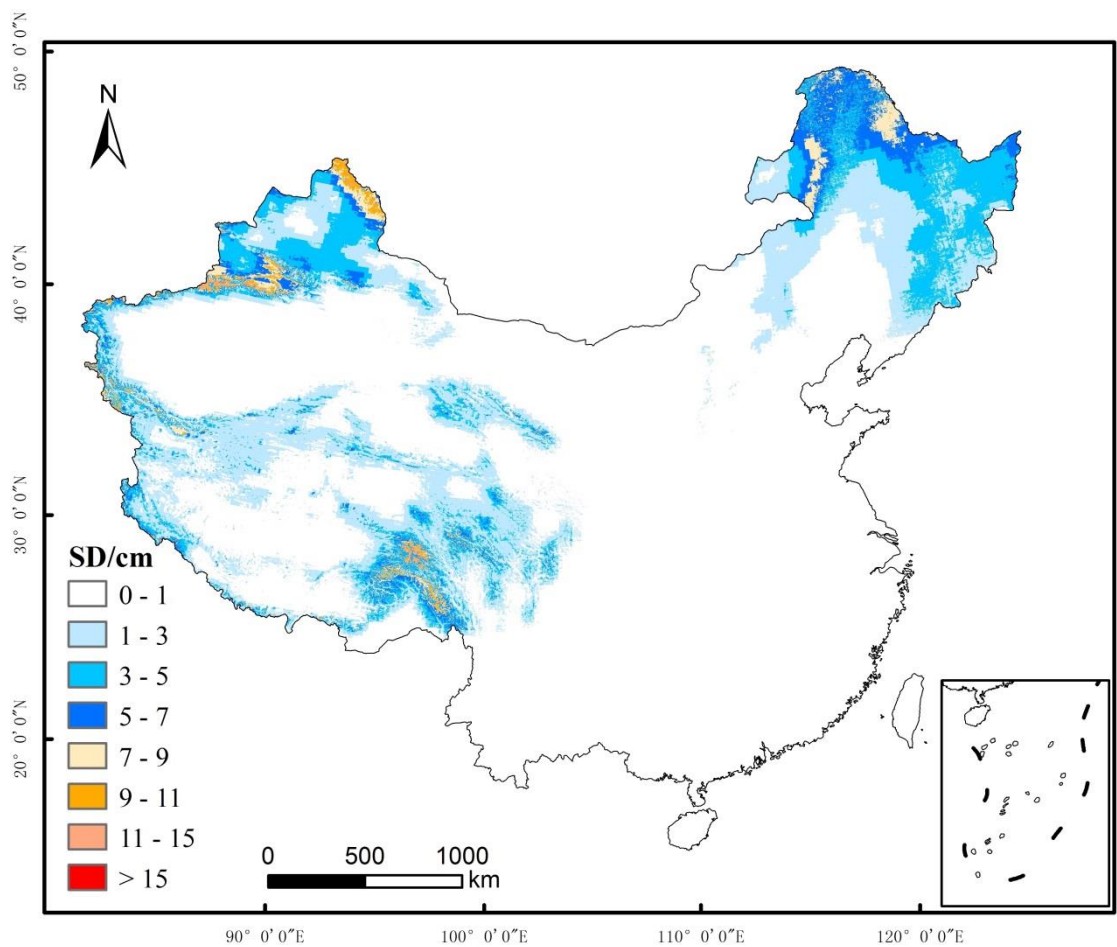

Figure 8: Spatial distribution of the average annual snow depth in China from December 2000 to
November 2014.

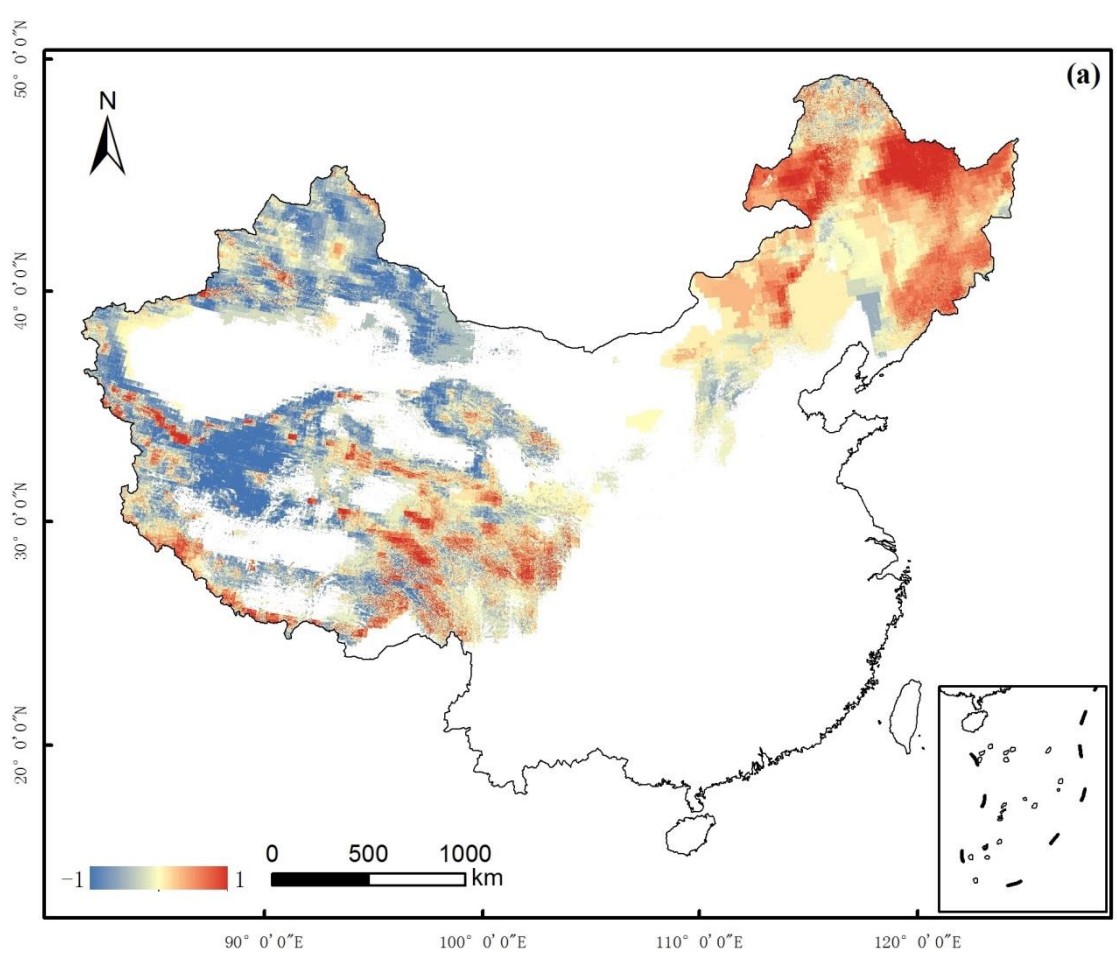

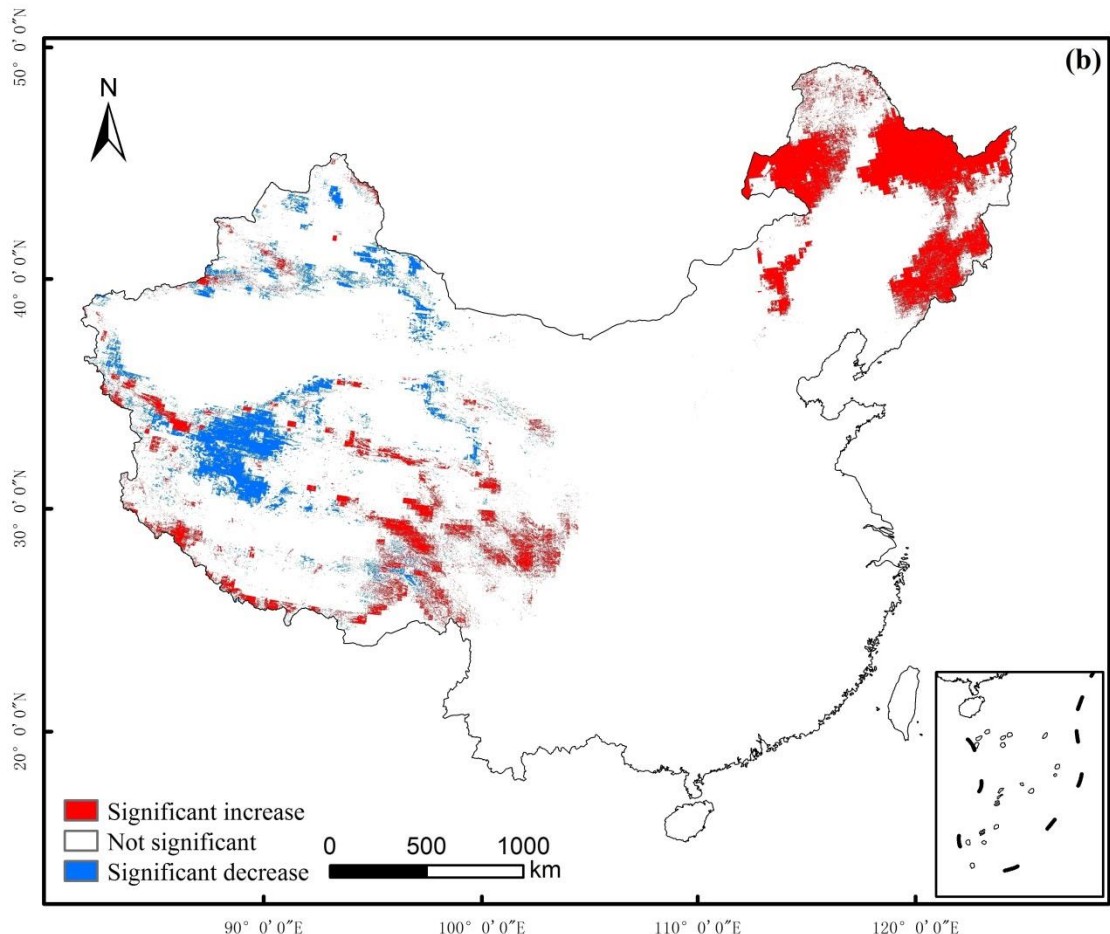

Figure 9: Variation in the average annual SD in China based on the Mann-Kendall method between
2001 and 2014. (a) Variation in the average annual SD; (b) significance of the variation in the average
annual SD.

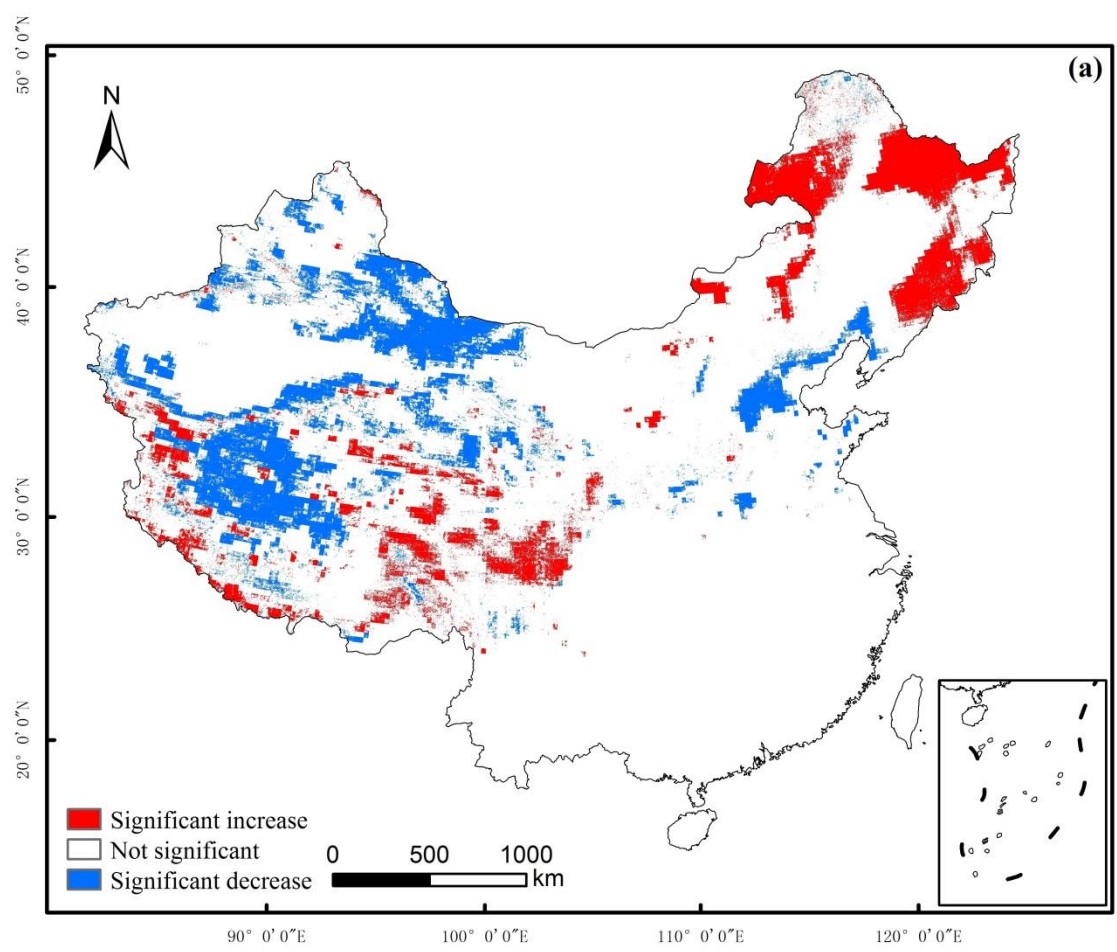

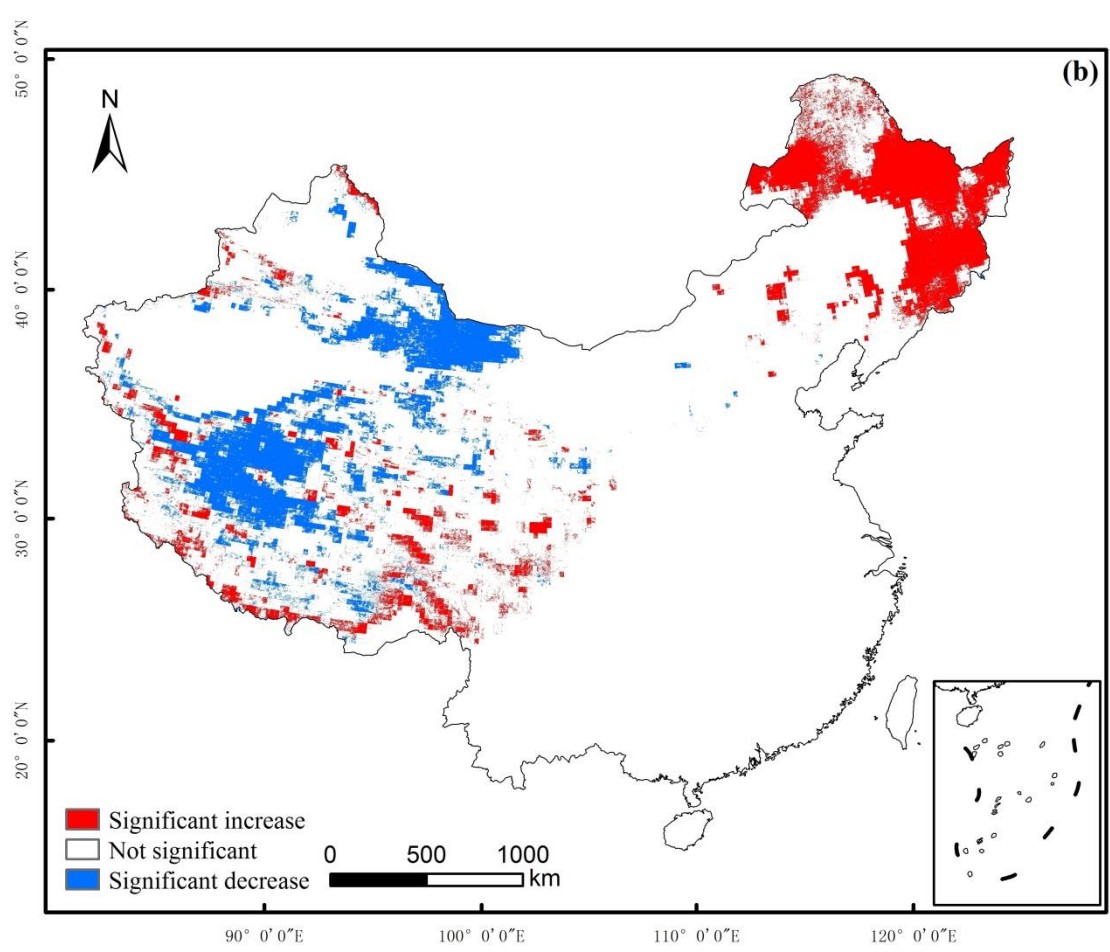

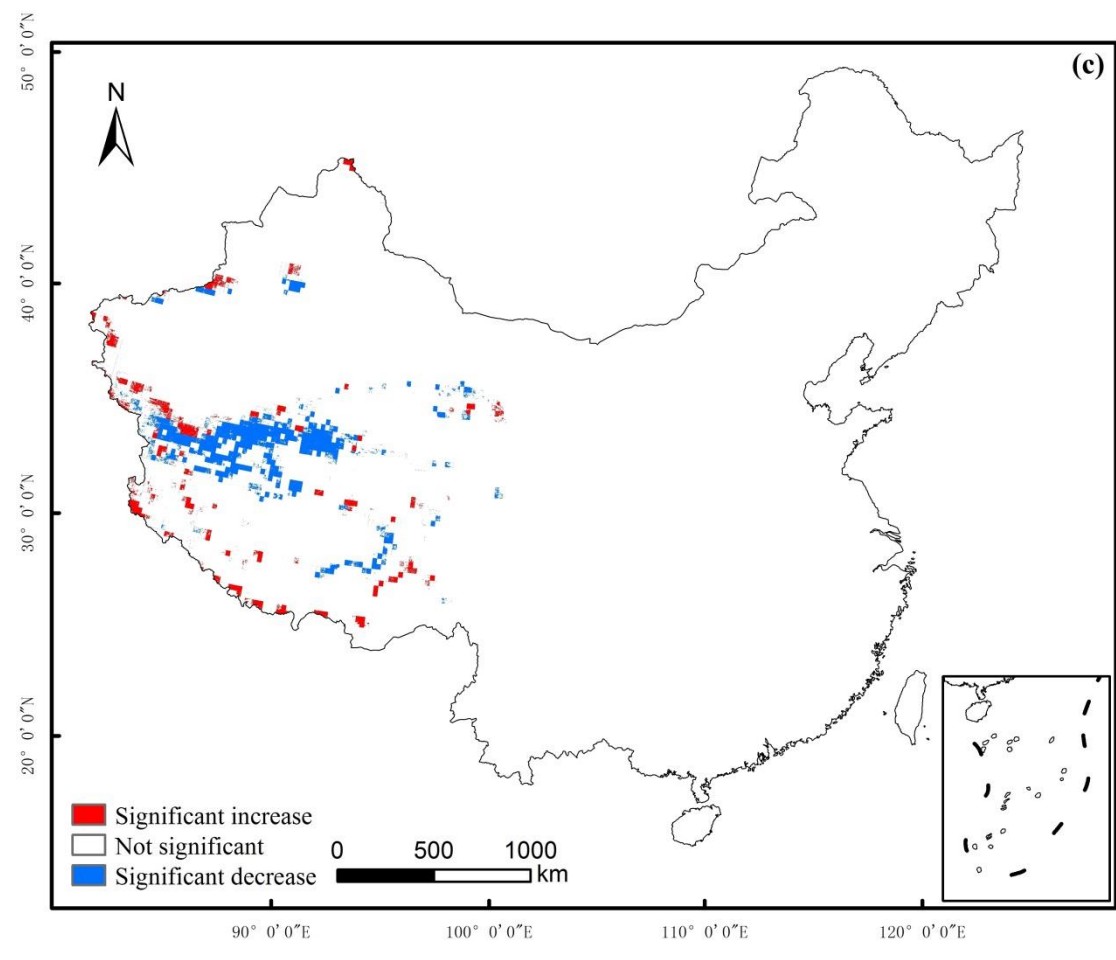

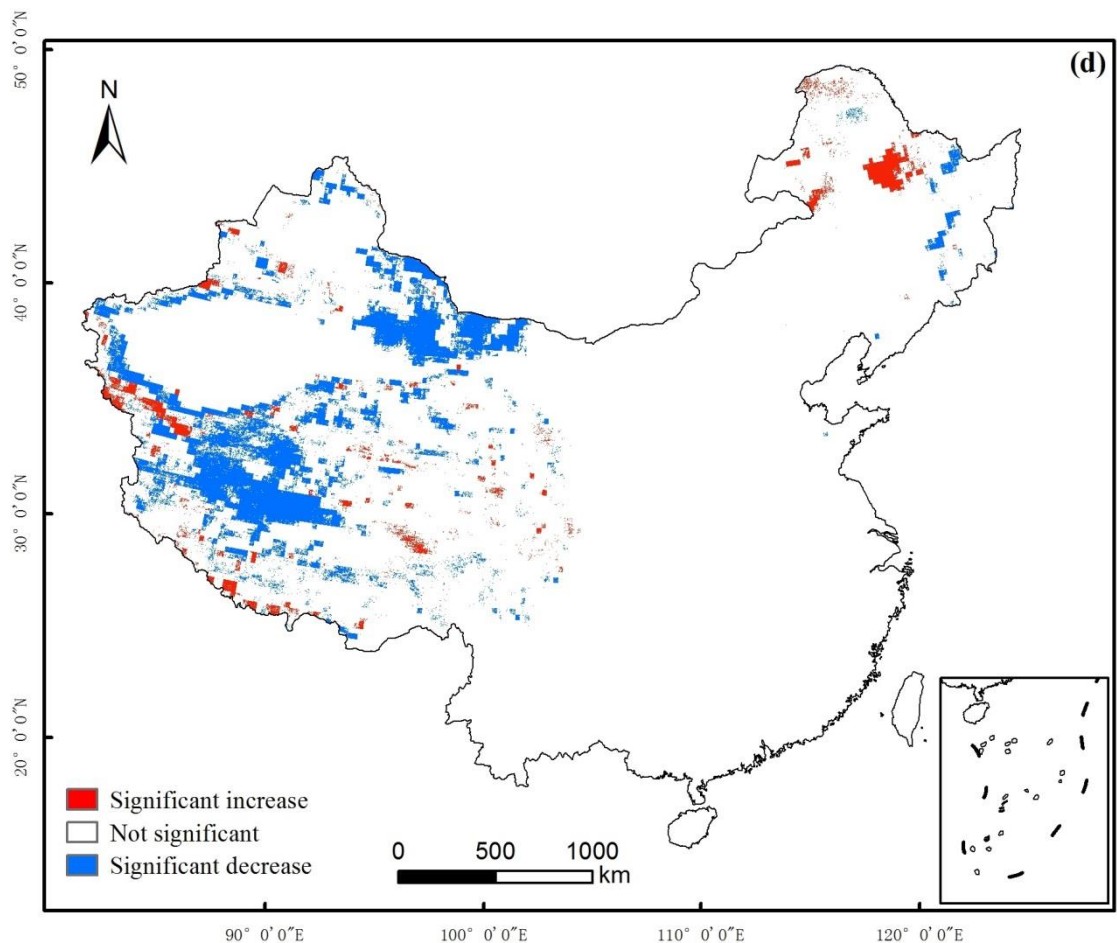

Figure 10: Variation in the average SD during each season in China based on the M-K method from 2001to 2014. (a), (b), (c), and (d) show the significance of the variations during the winter, spring, summer, and fall, respectively.