# Peer review of "Spatio-temporal dynamics of snow cover based on multi-source remote sensing data in China"

_The Cryosphere, 2016_

## Referee Comment (RC1) · Anonymous Referee #1 · 25 Jul 2016

The Huang et al paper used the combined MODIS snow cover and passive microwave snow depth data to produce a daily cloudless snow cover and 500 m snow depth (not daily based on the eqn 1), to analyze the snow cover day, snow cover area, and snow depth variation for China for the period of 2000-2014. They found the overall annual number of snow covered days increased (except in summer), average snow covered area did not change much (summer and winter decreased, spring and fall increased), and snow depth decreased (except in spring). They also analyze their spatial distribution of these changes and found snow cover significantly increased in south china and northeast China, but decreased in Xinjiang. Overall, I found the paper has some good results and may be publishable with carefully addressing my comments below. One of

my major comments is the English writing that needs to be carefully edited throughout the paper. The second one is the lack of discussion of their results with other published results, (such as the recent published Ke et al., 2016); without discussion, we do not know how this result differing from or similar as the known literature. The third one is the possible reason behind to all of the variation and changes. I know the last one is hard and I do not expect a thorough explanation, but some qualitative discussions are needed.

Below are some general comments: Abstract: the abstract writing is not very clear and needs to rewrite and more organized and more clarification. For example, in Line 15-19, they talked about the snow covered days and snow cover area, but these two contents are mixed in several sentences; snow depth is also mentioned here, but later in line 20-21, snow depth is mentioned again. I also confused in the line 15, they said snow depth increased, but 20-21, snow depth was decreased except in spring. Unless the increase in spring is much larger than the decreased in other seasons, it is not possible to see the annual snow depth was increased as stated in line 15. If this is the case, then authors should make this statement clear, not let readers to figure it out. Also the last sentence in the abstract, authors should say all regions with increase together, then all regions with decrease together, not as did here. Also it is not clear in the last sentence, snow cover means snow cover area, days, or depth?

1. Introduction: the part should be more focused on the topic of study and does not need to include everything that does not link to the topic of snow cover change in China. From 52 to 92, authors list many snow cover studies, I don't think it make sense, you should only mention the most relevant and should discuss in the end of the paper that how your results differ from, similar with or extent those studies, so your study is not just a study, but a significant addition to the current literature.

2.1 study area: you include "why your study is important" here, but you do not need to repeat here again and it should be in the introduction. It is clear based on the figure 1, you should basically talk more about the elevation distribution, a little bit about the
population distribution and economy, etc. . .

3. Results The current organization is very confusing and not easy to follow. You should reorganize the content into: snow covered days, snow covered area, snow depth. One at a time, not mixing them together.

4. you need a discussion section, to put your results in the big picture of literature, how your results differ from, similar as, or extent in certain degree of the current literature. You also need to include a paragraph on the possible explanations to the observed change, difference, or extension.

The paper needs a thorough English edits and I only catch a few below and will do a detailed comments after the first revision.

Line13, change "for December . . ." to "from the period of December. . ." L14, change "the snow cover" to "snow cover" L15, change "indicated" to "indicate" L140, by Dr. Huang, should be replace by "by Huang et al. 2014", L147, change "continent" to "land" L151-157, the equation is not clear, SDsp, what sp means here? The equation only give the annual snow depth for each snow pixel, right? Then make it clear here.

figures 4-8, captions, remove the "analysis result maps of the"

---

## Referee Comment (RC2) · Anonymous Referee #2 · 8 Aug 2016

In this manuscript, a synthesized snow cover product was produced first, which combined optical and passive remote sensing snow cover products. Cloud removal method and downscaling method were developed to retain the advantage of both optical and passive remote sending product, i.e., fine spatial resolution and cloudless, respectively. Then, based on the product, spatiotemporal dynamics of snow cover in China over the past 14 years were carefully analyzed. As a good data is the foundation of a reliable analysis. This synthesized snow cover product is considered of high quality, due to reasonable cloud removal and downscaling method. Also the analyses are well-organized, the results are quite specific. So, this manuscript is considered quite suitable to this journal. But still, some minor revisions are needed.

1. The descriptions of sentences need to be more carefully considered, especially some improper prepositions. In addition, some confused words or sentences are listed below: a) Line 31: "Middle-latitude". Usually we say middle latitude, or mid-latitude, but merely middle-latitude. b) Line 155: "SDi is the 25-km spatial resolution snow depth value in year i". This definition is not clear to me, as I cannot tell if SDi should be a daily result or annual mean result. c) Line 194: "Because some remote sensing data were lost". This sentence is quite confusing, especially with the word "lost". d) Line 268: "(December-February next year". There should be a ")" after "(" .

2. Some detailed problems in figures. a) Resolution of figures (dpi) should be enhanced, especially the maps. b) In figure titles, when it refers to "average annual", it is suggested to add time duration. Take Figure 3 for example, it is advised to be: "... annual average snow depth in China from 2001to 2014".

3. There are some strange "missing" words or blanks throughout the manuscript. a) Line 173, Line 177: "at a given significance level " b) Line 233: "( <0) "

4. There are some leap years during the study period, but it seems that you assumed every year to be 365 days. Explanations are needed.

5. As you speak highly of the M-K method in analyzing the variation and trend of snow cover data, why you used Sen's median method "to test the accuracy of this result" (Line 230)? Do you have any explanations?

6. The long time series of snow depth in China you used in WESTDC have been updated based on the following publications:

a) Che, T., Dai, L.Y., Zheng, X.M., Li, X.F., Zhao, K., 2016. Estimation of snow depth from MWRI and AMSR-E data in forest regions of Northeast China. Remote Sensing of Environment 183, 334-349. b) Dai, L., Che, T., Ding, Y., 2015. Inter-Calibrating SMMR, SSM/I and SSMI/S Data to Improve the Consistency of Snow-Depth Products in China. Remote Sensing 7, 7212. c) Dai, L.Y., Che, T., Wang, J., Zhang, P., 2012.

[Figure]

Snow depth and snow water equivalent estimation from AMSR-E data based on a priori snow characteristics in Xinjiang, China. Remote Sensing of Environment 127, 14-29.

---

## Author Comment (AC1) · 5 Sep 2016

The Huang et al paper used the combined MODIS snow cover and passive microwave snow depth data to produce a daily cloudless snow cover and 500 m snow depth (not daily based on the eqn 1), to analyze the snow cover day, snow cover area, and snow depth variation for China for the period of 2000-2014. They found the overall annual number of snow covered days increased (except in summer), average snow covered area did not change much (summer and winter decreased, spring and fall increased), and snow depth decreased (except in spring). They also analyze their spatial distribution of these changes and found snow cover significantly increased in south china and northeast China, but decreased in Xinjiang. Overall, I found the paper has some good

results and may be publishable with carefully addressing my comments below. One of my major comments is the English writing that needs to be carefully edited throughout the paper. The second one is the lack of discussion of their results with other published results, (such as the recent published Ke et al., 2016); without discussion, we do not know how this result differing from or similar as the known literature. The third one is the possible reason behind to all of the variation and changes. I know the last one is hard and I do not expect a thorough explanation, but some qualitative discussions are needed.

Response: Firstly, on behalf of all authors, we want to thank you for your affirmation for our work, and also your great help and suggestions for this manuscript. We are sorry the inconvenience caused to you about our English writing. The editorial changes for language usage throughout were made by a native English scientific editor in this version. In addition, based on your suggestion, a discussion section was added in the manuscript. We compared the Ke's as well as others results with our conclusions in discussion, and the possible reason behind the snow variation was also discussed. Thank you again for the great suggestion.

Author's changes in manuscript: Please see our responses to the above comments in revised manuscript.

Below are some general comments: Abstract: the abstract writing is not very clear and needs to rewrite and more organized and more clarification. For example, in Line 15-19, they talked about the snow covered days and snow cover area, but these two contents are mixed in several sentences; snow depth is also mentioned here, but later in line 20-21, snow depth is mentioned again. I also confused in the line 15, they said snow depth increased, but 20-21, snow depth was decreased except in spring. Unless the increase in spring is much larger than the decreased in other seasons, it is not possible to see the annual snow depth was increased as stated in line 15. If this is the case, then authors should make this statement clear, not let readers to figure it out. Also the last sentence in the abstract, authors should say all regions with increase

together, then all regions with decrease together, not as did here. Also it is not clear in the last sentence, snow cover means snow cover area, days, or depth?

Response: We rewrote the Abstract based on your suggestion. Thanks.

Author's changes in manuscript: Abstract. By combining optical remote sensing snow cover products with passive microwave remote sensing snow depth (SD) data, we produced a MODIS cloudless binary snow cover product and a 500 m spatial resolution downscaling snow depth product. The temporal and spatial variations of snow cover from December 2000 to November 2014 in China were analyzed. The results indicate that over the past 14 years, 1) the perennial average annual snow-covered area (SCA) in China has been 11.3% over the entire year and 27% during the winter. The average SCA during the summer and winter decreased, whereas the average SCA during the spring and fall increased, and the overall average annual SCA did not change significantly. 2) The snow-covered days (SCDs) during the winter, spring, and fall all increased, whereas they decreased during the summer, and the annual SCDs increased. 3) The average SD during the winter, summer, and fall decreased, the average SD during the spring increased, and the overall average SD increased over the past decade. 4) The spatial distributions of the areas of increase and decrease average annual SD were highly consistent with those of the annual SCDs, and were also consistent with the variations in each season. 5) The regional differences in the variation of snow cover in China were significant. The SCD and SD increased significantly in South and Northeast China, decreased significantly in northern Xinjiang Province. The SCD and SD increased on the southwest edge and in the southeast part of the Tibetan Plateau, whereas it decreased in the north and northwest regions.

1. Introduction: the part should be more focused on the topic of study and does not need to include everything that does not link to the topic of snow cover change in China. From 52 to 92, authors list many snow cover studies, I don't think it make sense, you should only mention the most relevant and should discuss in the end of the paper that how your results differ from, similar with or extent those studies, so your study is not

just a study, but a significant addition to the current literature.

Response: We revised introduction part based on your comments, Thanks. First, the references focused on the snow cover change in China were moved in discussion part, which as a basis to compare with our results. Second, the origin introduction of the study area was also moved in Introduction, to emphasis the important of this study. Author's changes in manuscript: 1 Introduction. Snow cover is closely related to human lives, and it has both positive and negative effects (Liang et al., 2004). High and mid-latitude regions contain abundant snow cover and glacial resources, which are the source regions for many rivers (Zhang et al., 2002). Snowmelt runoff can make up more than 50% of the total discharge of many drainage basins (Seidel and Martinec, 2004). Snow cover is an important resource for industrial, agricultural, and domestic water use. Especially in arid and semi-arid regions, the development of agricultural irrigation and animal husbandry relies on the melting of snow cover (Pulliainen, 2006; Li, 2001). Winter water deficiencies can easily cause droughts (Cezar Kongoli et al., 2012). On the other hand, flood disasters caused by melting snow cover and snow disasters such as avalanches, glacial landslides, and snowdrifts are also common (Gao et al., 2008; Liu et al., 2011; Shen et al., 2013). Rising temperatures due to global warming rapidly change the snow cover conditions in seasonal snow-covered regions, which has led to accelerated melting of most ice sheets and permanent snow covers (Yao et al., 2012), increasing snowline elevations (Chen, 2014), decreasing wetland areas, and the reallocation of precipitation, which has further led to frequent floods and snow disasters (Lee et al., 2013; Wang et al., 2013). Global warming is an indisputable fact. Rising temperature will strongly affect alpine and polar snow cover (IPCC, 2013). The variation of global and regional snow covers greatly affects the use of snow resources by humanity, and the feedback mechanism of albedo further affects climate (Bloch, 1964; Robinson, 1997; Nolin and Stoeve, 1997). Several studies have indicated that the snow cover in the alpine regions in China affect the atmospheric circulation and weather systems in East Asia and further affect the climate in China (Qian et al., 2003; Zhao et al., 2007). Alpine snow cover has important implications for hydrology, climate, and the ecological environment (Chen and Liu, 2000; Hahn and Shula, 1976). China is large, and its snow-covered regions are widely distributed geographically. North Xinjiang, Northeast China-Inner Mongolia, and the Tibetan Plateau are the 3 major regions with seasonal snow cover in China (Wang et al., 2009). They are also the major pasturing regions. Winter and spring snowfalls are the major water resources in north Xinjiang and the Tibetan Plateau (Pei et al., 2008; Chen et al., 1991; Wang et al., 2014). Heavy snowfall can also cause severe snow disasters and large numbers of livestock deaths (Liu et al., 2008; Chen et al., 1996). Floods caused by melting snow cover also frequently occur in the spring, severely limit the development of grassland animal husbandry and affect the safety of human lives (Shen et al., 2013). Therefore, accurate acquisition of snow-covered area (SCA) and SD information is significant for understanding climate change and the hydrological cycle, conducting water resource surveys, and preventing and forecasting snow disasters in China. Recent studies of the distribution and variation of snow cover in China have progressed greatly, but they have mainly focused on the Tibetan Plateau, Xinjiang, and Northeast China (Chen and Li, 2011). Furthermore, the results from different snow cover datasets are slightly different, and the snow cover variations in different regions are also different. MODIS data, which have high spatial and temporal resolution, have been widely used in the remote sensing fields of ecology, atmospheric science, and hydrology. However, clouds strongly interfere with optical sensors. Hence, we cannot directly use snow cover products acquired by optical sensors to effectively quantify SCA. Passive microwaves can penetrate clouds and are not affected by weather. However, the coarse resolution of passive microwave products greatly limits the accuracy of regional snow cover monitoring. Therefore, cloud removal and downscaling are effective approaches for enhancing the accuracy of snow cover monitoring using optical and passive microwave products, respectively. This study used the MODIS daily snow cover product and passive microwave SD data to produce a daily cloudless SCA product and a downscaled SD product with a 500 m spatial resolution. The integrated daily snow products are used to analyze the temporal and spatial variations of the snow cover in China from December 2000 to November

2014 and quantitatively evaluate the variation of SCA, snow-covered days (SCDs), and average SD to provide a basis for further understanding the interaction between climate and snow cover under the background of globe warming in China.

2.1 study area: you include "why your study is important" here, but you do not need to repeat here again and it should be in the introduction. It is clear based on the figure 1, you should basically talk more about the elevation distribution, a little bit about the population distribution and economy, etc: : :

Response: Did as you suggested. We moved this paragraph in the introduction, and rewrote the study area based on your suggestion. Thanks a lot. Author's changes in manuscript: 2.1 Study area. China has a large area and a large population. Its relief is high in the west and low in the east. Mountains, plateaus, and hills account for approximately 67% of the land area, and basins and plains account for approximately 33%. The mountains are mostly oriented east-west and northeast-southwest, including the Altun Mountains, Tianshan, Kunlun, Karakoram, Himalaya, Yinshan, Qinling, Nanling, Daxing'anling, Changbaishan, Taihang, Wuyi, Taiwan, and Hengduan. The Tibetan Plateau, which has an average elevation of more than 4000 m, is located to the west and is known as the "Roof of the World". Mount Everest is 8844.43 m tall and is the highest mountain in the world. To the north and east, Inner Mongolia, the Xinjiang area, the Loess Plateau, the Sichuan Basin, and the Yunnan-Guizhou Plateau are second-stage terrains of China. The region from east of the Daxing'anling-Taihang-Wushan-Wuling-Xuefeng Mountains to the shoreline mostly contains third-stage terrains composed of plains and hills with an average elevation of less than 1000 m. The multi-year stable snow cover is mainly distributed in the Tibetan Plateau, Northeast China and Inner Mongolia, and northern Xinjiang and covers a total area of 4,200,000 km2. This snow cover forms the major reservoirs for the seasonal snow-cover water resources in China (Li et al., 1983).

3. Results: The current organization is very confusing and not easy to follow. You should reorganize the content into: snow covered days, snow covered area, snow

depth. One at a time, not mixing them together.

Response: We re-organized the Results as you suggested. The SCA, SCD, and the SD were separated as three parts of results in the revised manuscript. We fully agreed with your suggestion, thanks a lot. Author's changes in manuscript: The structure of the revised manuscript is as follows: 1 Introduction 2 Materials and Methods 2.1 Study area 2.2 Remote sensing snow products 2.3 Cloud removal and downscaling algorithms 2.4 Analysis of the snow cover variation 3 Results 3.1 Snow-Covered Area 3.2 Snow-Covered Days 3.3 Snow Depth 4 Discussion 5 Conclusion

4. You need a discussion section, to put your results in the big picture of literature, how your results differ from, similar as, or extent in certain degree of the current literature. You also need to include a paragraph on the possible explanations to the observed change, difference, or extension.

Response: A discussion section was added this time. Author's changes in manuscript: 4 Discussions Snow cover is widely distributed in China. Researchers have conducted numerous studies on the variation of the snow cover in China using remote-sensing snow cover data from various satellites. By combining the MODIS Terra and Aqua snow cover data, Liu et al. (2012) studied the spatial stability of the three major snow-covered regions in China for 2001–2010 and analyzed the characteristics of the seasonal and annual snow cover variations. The results indicated that the snow cover stabilities in the three major snow-covered regions were in the order of Xingjiang > Northeast China-Inner Mongolia > Tibetan Plateau. The stable SCA in China did not change significantly. Wang et al. (2012) used combined MODIS and AMSR-E data to obtain a cloudless snow cover product and analyzed the temporal and spatial distributions of the snow cover in the arid regions in China for 2002–2009. The results indicated that the SCA in the stable snow-covered regions did not change significantly, whereas the unstable snow-covered regions had large annual variations in the SCA. The results of this study indicated that the average annual SCA did not change significantly. The stable snow-covered regions in China ($60 < SCD \leq 350$) were primarily

located in Northeast China-Inner Mongolia, north Xinjiang, and the high mountains in the Tibetan Plateau, and the stable snow area did not change significantly during 2001-2014. Che et al. (2005) used the SD data that were inverted from SSM/I passive microwave data to analyze the snow cover distribution and variations in China for 1993–2002. The results indicated that the snow cover reservoir in China did not increase or decrease significantly over that ten-year period. The winter snow cover reservoir was mainly located in the three major stable snow-covered regions of Xinjiang, the Tibetan Plateau, and Northeast China. Dai et al. (2010) indicated that the number of SCDs and the SD in China increased between 1978 and 2005. The western Tibetan Plateau was a sensitive region with an abnormal variation in SCDs, whereas north Xinjiang, the mountainous regions in Northeast China and the east-central Tibetan Plateau were sensitive regions with abnormal SD variations. Dou et al. (2010) used the MODIS snow cover product to study the Tianshan Mountains in China, and indicated that the SCA in the Tianshan Mountains increased slightly; the increase was especially significant in the winter. Furthermore, the snow cover decreased in the regions at with elevation of $\geq$ 4000 m and increased in the regions with elevation of < 4000 m. This study found similar results, but the significant increase in SCDs was observed in the spring, not in the winter. Wang et al. (2015) used the MODIS cloudless synthesized snow cover product to further study the Tibetan Plateau and showed that the maximum SCDs and the perennial SCA in the plateau region decreased, the SCA increased, and the variations in temperature and precipitation significantly affected the plateau snow cover. The study by Basang et al. (2012) on the variation of snow cover in Tibet indicated that from 1980 to 2009, the SCDs and maximum SD in Tibet decreased. The decrease was very significant after the start of the 21th century. The variations were slightly different in different seasons, and the results observed by different remote-sensing satellites were also different. The NOAA data showed that the snow cover decreased during the winter and summer and increased during the spring and fall. The MODIS data showed that the snow cover decreased during the summer, spring, and winter, and only increased during the fall. Our study showed that over the past 14 years, the SCDs

and SD decreased primarily in the hinterlands of the Tibetan Plateau, and increased in the southwest and southeast margins of the Tibetan Plateau. Studies based on long time series of observations by ground stations have indicated that the number of SCDs and the SD in Northeast China increased every year (Chen and Li, 2011; Yan et al., 2015; Ke et al., 2016), which is consistent with our results for Northeast China over the past 14 years. The response of snow cover to global climate change has always been a popular topic among researchers in China and abroad. Aided by high-temporal-resolution optical and passive microwave remote-sensing data, researchers can simply and rapidly monitor the dynamic changes of snow cover over long periods of time. Previous studies have indicated that low-elevation regions were susceptible to the influence of precipitation, whereas high-elevation regions were more susceptible to the influence of temperature (Xu et al., 2007). As temperatures rise, precipitation increases, which leads to accelerations of snow melting rates in high-elevation regions and a decrease in the SCA. However, this pattern causes more moisture to participate in the atmospheric water cycle, which increases the precipitation in low-elevation regions and further increases the SCA. Thus, the snow cover increased significantly in South and Northeast China, and increased on the southwest margin and in the southeast region of the Tibetan Plateau, whereas the SCA in the north and northwest parts of the Tibetan Plateau generally decreased. However, Hu et al. (2013) indicated that the snow cover in north Xinjiang exhibited a good negative correlation with temperature but an insignificant correlation with precipitation. Therefore, the snow cover in Xinjiang has decreased overall with global warming.

The paper needs a thorough English edits and I only catch a few below and will do a detailed comments after the first revision. Line13, change "for December : : :" to "from the period of December: : :" L14, change "the snow cover" to "snow cover" L15, change "indicated" to "indicate" L140, by Dr. Huang, should be replace by "by Huang et al. 2014", L147, change "continent" to "land" L151-157, the equation is not clear, SDsp, what sp means here? The equation only give the annual snow depth for each snow pixel, right? Then make it clear here. figures 4-8, captions, remove the "analysis

result maps of the"

Response: Revised based on your comments. Thank you. Author's changes in manuscript: SDsp is the sub-pixel daily SD with a 500 m spatial resolution, SD is the daily SD with a 25 km spatial resolution, SDYi is the average number of SCDs for each MODIS pixel in year i, and SDTi is the sum of the total SCDs for each SD pixel in year i. Other responses to your comments please see the revised manuscript.

New references updated are as follows: Che, T., Dai, L.Y., Zheng, X.M., Li, X.F., Zhao, K.: Estimation of snow depth from MWRI and AMSR-E data in forest regions of Northeast China. Remote Sens. Environ, 183, 334-349, 2016. Dai, L., Che, T., Ding, Y.: Inter-Calibrating SMMR, SSM/I and SSMI/S Data to Improve the Consistency of Snow-Depth Products in China. Remote Sens, 7, 7212-7230, 2015. Dai, L.Y., Che, T., Wang, J., Zhang, P.: Snow depth and snow water equivalent estimation from AMSR-E data based on a priori snow characteristics in Xinjiang, China. Remote Sens. Environ, 127, 14-29, 2012. Hall, D. K., Riggs, G. A., Salomonson, V. V., Digirolamo, N. E., & Bayr, K. J.: MODIS snow-cover products. Remote Sens. Environ, 83(1): 181-194, 2002. Qian, Y. F., Zheng, Y. Q., Zhang, Y., Miao, M. Q.: Responses of China's summer monsoon climate to snow anomaly over the Tibetan Plateau. Int. J. of Climatol., 23, 593-613, 2003. Zhao, P., Zhou, Z. J., Liu, J. P.: Variability of the Tibetan spring snow and its associations with the hemispheric extropical circulation and East Asian summer monsoon rainfall: An observational investigation. J. Climate, 20, 3942-3955, 2007. Ke, C. Q., Li, X. C., Xie, H. J., Ma, D. H., Liu, X., Kou, C.: Variability in snow cover phenology in China from 1952 to 2010. Hydrol. Earth Syst. Sci., 20, 755-770, 2016.

Please also note the supplement to this comment:
http://www.the-cryosphere-discuss.net/tc-2016-124/tc-2016-124-AC1-supplement.pdf

---

## Author Response (AR1)

- 5 (1) Comments from Referees: The Huang et al paper used the combined MODIS snow cover and passive microwave snow depth data to produce a daily cloudless snow cover and 500 m snow depth (not daily based on the eqn 1), to analyze the snow cover day, snow cover area, and snow depth variation for China for the period of 2000-2014. They found the overall annual number of snow covered days increased (except in summer), average snow covered area did not change much (summer and winter decreased, spring and fall increased), and snow
- depth decreased (except in spring). They also analyze their spatial distribution of these changes and found snow cover significantly increased in south china and northeast China, but decreased in Xinjiang. Overall, I found the paper has some good results and may be publishable with carefully addressing my comments below. One of my major comments is the
- 15 English writing that needs to be carefully edited throughout the paper. The second one is the lack of discussion of their results with other published results, (such as the recent published Ke et al., 2016); without discussion, we do not know how this result differing from or similar as the known literature. The third one is the possible reason behind to all of the variation and changes. I know the last one is hard and I do not expect a thorough explanation, but some
- 20 qualitative discussions are needed.

Author's Response: Firstly, on behalf of all authors, we want to thank you for your affirmation for our work, and also your great help and suggestions for this manuscript. We are sorry the inconvenience caused to you about our English writing. The editorial changes for language usage throughout were made by a native English scientific editor in this version. In addition,

- 25 based on your suggestion, a discussion section was added in the manuscript. We compared the Ke's as well as others results with our conclusions in discussion, and the possible reason behind the snow variation was also discussed. Thank you again for the great suggestion. Author's changes in manuscript: Please see our responses to the above comments in revised manuscript.
- 30 Below are some general comments:

(2) Comments from Referees: Abstract: the abstract writing is not very clear and needs to rewrite and more organized and more clarification. For example, in Line 15-19, they talked about the snow covered days and snow cover area, but these two contents are mixed in several sentences; snow depth is also mentioned here, but later in line 20-21, snow depth is mentioned again. I also confused in the line 15, they said snow depth increased, but 20-21, snow depth was decreased except in spring. Unless the increase in spring is much larger than

the decreased in other seasons, it is not possible to see the annual snow depth was increased as stated in line 15. If this is the case, then authors should make this statement clear, not let readers to figure it out. Also the last sentence in the abstract, authors should say all regions
with increase together, then all regions with decrease together, not as did here. Also it is not clear in the last sentence, snow cover means snow cover area, days, or depth?

Author's Response: We rewrote the Abstract based on your suggestion. Thanks.

**Author's changes in manuscript:**

35

Abstract. By combining optical remote sensing snow cover products with passive microwave

remote sensing snow depth (SD) data, we produced a MODIS cloudless binary snow cover 45 product and a 500 m snow depth product. The temporal and spatial variations of snow cover from December 2000 to November 2014 in China were analyzed. The results indicate that, over the past 14 years, (1) the mean snow-covered area (SCA) in China was 11.3% annually and 27% in winter season, with the mean SCA decreasing in summer and winter seasons, in 50 increasing in spring and fall seasons, and no much change annually; (2) the snow-covered days (SCDs) showed increasing in winter, spring, and fall, and annually, whereas decreasing in summer; (3) the average SD decreased in winter, summer, and fall, while increased in spring and annually; (4) the spatial distributions of SD and SCD were highly correlated seasonally and annually; and (5) the regional differences in the variation of snow cover in 55 China were significant. Overall, the SCD and SD increased significantly in South and Northeast China, decreased significantly in northern Xinjiang Province. The SCD and SD increased on the southwest edge and in the southeast part of the Tibetan Plateau, whereas it

decreased in the north and northwest regions.

(3) Comments from Referees: 1. Introduction: the part should be more focused on the topic ofstudy and does not need to include everything that does not link to the topic of snow cover

change in China. From 52 to 92, authors list many snow cover studies, I don't think it make sense, you should only mention the most relevant and should discuss in the end of the paper that how your results differ from, similar with or extent those studies, so your study is not just a study, but a significant addition to the current literature.

65 Author's Response: We revised introduction part based on your comments, Thanks. First, the references focused on the snow cover change in China were moved in discussion part, which as a basis to compare with our results. Second, the origin introduction of the study area was also moved in Introduction, to emphasis the important of this study.

**Author's changes in manuscript:**

[revised manuscript text omitted]

(4) Comments from Referees: 2.1 study area: you include "why your study is important" here,
but you do not need to repeat here again and it should be in the introduction. It is clear based on the figure 1, you should basically talk more about the elevation distribution, a little bit about the population distribution and economy, etc: : :

Author's Response: Did as you suggested. We moved this paragraph in the introduction, and rewrote the study area based on your suggestion. Thanks a lot.

130 Author's changes in manuscript:

2.1 **Study area**. China has a large area and a large population, with mountains, plateaus, and hills accounting for ~67% of the land area, and basins and plains for ~33% (Figure 1). The mountains are mostly oriented east-west and northeast-southwest, including the Altun Mountains, Tianshan, Kunlun, Karakoram, Himalaya, Yinshan, Qinling, Nanling,

- 135 Daxing'anling, Changbaishan, Taihang, Wuyi, Taiwan, and Hengduan. The Tibetan Plateau, which has an average elevation of more than 4000 m, is located to the southwest and is known as the "Roof of the World". Mount Everest is 8844.43 m in height and is the highest mountain in the world. To the north and east, Inner Mongolia, the Xinjiang area, the Loess Plateau, the Sichuan Basin, and the Yunnan-Guizhou Plateau are second-stage terrains of China. The
- region from east of the Daxing'anling-Taihang-Wushan-Wuling-Xuefeng Mountains to the shoreline mostly contains third-stage terrains composed of plains and hills with an average elevation of less than 1000 m. The multi-year stable snow cover is mainly distributed in the Tibetan Plateau, Northeast China and Inner Mongolia, and northern Xinjiang covering a total area of ~4,200,000 km2. This snow cover forms the major freshwater reservoirs for most part of China (Li et al., 1983).

(5) Comments from Referees:3. Results: The current organization is very confusing and not easy to follow. You should reorganize the content into: snow covered days, snow covered area, snow depth. One at a time, not mixing them together.

Author's Response: We re-organized the Results as you suggested. The SCA, SCD, and theSD were separated as three parts of results in the revised manuscript. We fully agreed with

your suggestion, thanks a lot.

Author's changes in manuscript:

The structure of the revised manuscript is as follows:

1 Introduction

**155 2 Materials and Methods**

- 2.1 Study area
- 2.2 Remote sensing snow products
- 2.3 Cloud removal and downscaling algorithms
- 2.4 Analysis of the snow cover variation

**160 3 Results**

- 3.1 Snow-Covered Area
- 3.2 Snow-Covered Days
- 3.3 Snow Depth

4 Discussion

**165 5 Conclusion**

(6) Comments from Referees: 4. You need a discussion section, to put your results in the big picture of literature, how your results differ from, similar as, or extent in certain degree of the current literature. You also need to include a paragraph on the possible explanations to the observed change, difference, or extension.

**170 Author's Response: A discussion section was added this time.**

Author's changes in manuscript:

4 Discussions

Snow cover is widely distributed in China. The results of this study indicated that the average annual SCA did not change significantly. The relative stable snow-covered regions (60 < SCD

- 175  $\leq$  350) in China were primarily located in Northeast China-Inner Mongolia, north Xinjiang, and the high mountains in the Tibetan Plateau, and the stable snow area did not change significantly during 2001-2014. Liu et al. (2012) studied the spatial stability of the three major snow-covered regions in China for 2001–2010 and analyzed the characteristics of the seasonal and annual snow cover variations. The results indicated that the snow cover
- 180

China-Inner Mongolia > Tibetan Plateau. The stable SCA in China did not change significantly. Same results also found for the relative stable snow-covered regions, whereas the unstable snow-covered regions (SCDs < 60) had large annual variations in the SCA (Wang et al. 2012). Dou et al. (2010) used the MODIS snow cover product to study the Tianshan Mountains in China, and indicated that the snow cover in the Tianshan Mountains

- 185 Tianshan Mountains in China, and indicated that the snow cover in the Tianshan Mountains increased slightly; the increase was especially significant in the winter. Furthermore, the snow cover decreased in the regions at with elevation of  $\geq$  4000 m and increased in the regions with elevation of < 4000 m. This study found similar results, but the significant increase in SCDs was observed in the spring, not in the winter.
- 190 Dai et al. (2010) indicated that the number of SCDs and the SD in China increased between 1978 and 2005. The western Tibetan Plateau was a sensitive region with an abnormal variation in SCDs, whereas north Xinjiang, the mountainous regions in Northeast China and the east-central Tibetan Plateau were sensitive regions with abnormal SD variations. Che et al. (2005) used the SD data that were inverted from SSM/I passive microwave data to analyze
- 195 the snow cover distribution and variations in China for 1993–2002. The results indicated that the snow cover reservoir in China did not increase or decrease significantly over that ten-year period. The winter snow cover reservoir was mainly located in the three major stable snow-covered regions of Xinjiang, the Tibetan Plateau, and Northeast China. The study by Basang et al. (2012) on the variation of snow cover in Tibet indicated that from 1980– to 2009,
- the SCDs and maximum SD in Tibet decreased. The decrease was very significant after the start of the 21th century. The variations were slightly different in different seasons, and the results observed by different remote-sensing satellites were also different. Our study showed that over the past 14 years, the SCDs and SD decreased primarily in the hinterlands of the Tibetan Plateau, and increased in the southwest and southeast margins of the Tibetan Plateau.
  Studies based on long time series of observations by ground stations have indicated that the
- number of SCDs and the SD in Northeast China increased every year (Chen and Li, 2011; Yan et al., 2015; Ke et al., 2016), which is consistent with our results for Northeast China over the past 14 years.

(7) Comments from Referees: The paper needs a thorough English edits and I only catch afew below and will do a detailed comments after the first revision. Line13, change "for

December : : : " to "from the period of December: : : " L14, change "the snow cover" to "snow cover" L15, change "indicated" to "indicate" L140, by Dr. Huang, should be replace by "by Huang et al. 2014", L147, change "continent" to "land" L151-157, the equation is not clear, SDsp, what sp means here? The equation only give the annual snow depth for each snow

215 pixel, right? Then make it clear here. figures 4-8, captions, remove the "analysis result maps of the"

Author's Response: Revised based on your comments. Thank you.

Author's changes in manuscript:  $SD_{sp}$  is the sub-pixel daily SD with a 500 m spatial resolution, *SD* is the daily SD with a 25 km spatial resolution, *SDYi* is the average number of SCDs for

each MODIS pixel in year *i*, and *SDTi* is the sum of the total SCDs for each SD pixel in year i.Other responses to your comments please see the revised manuscript.
(1) Comments from Referees: In this manuscript, a synthesized snow cover product was produced first, which combined optical and passive remote sensing snow cover products. Cloud removal method and downscaling method were developed to retain the advantage of

- both optical and passive remote sending product, i.e., fine spatial resolution and cloudless, respectively. Then, based on the product, spatiotemporal dynamics of snow cover in China over the past 14 years were carefully analyzed. As a good data is the foundation of a reliable analysis. This synthesized snow cover product is considered of high quality, due to reasonable cloud removal and downscaling method. Also the analyses are well-organized, the results are quite specific. So, this manuscript is considered quite suitable to this journal. But still, some minor revisions are needed.
  - 1. The descriptions of sentences need to be more carefully considered, especially some improper prepositions. In addition, some confused words or sentences are listed below: a) Line 31: "Middle-latitude". Usually we say middle latitude, or mid-latitude, but merely
- 240 middle-latitude. b) Line 155: "SDi is the 25-km spatial resolution snow depth value in year i". This definition is not clear to me, as I cannot tell if SDi should be a daily result or annual mean result. c) Line 194: "Because some remote sensing data were lost". This sentence is quite confusing, especially with the word "lost". d) Line 268: "(December-February next year". There should be a ")" after "(".
- 245 Author's Response: We are sorry the inconvenience caused to you about our English writing. Some confusing sentences you mentioned were revised. In additional, editorial changes for language usage throughout were also made by a native English scientific editor.

Author's changes in manuscript:

- a) High and mid-latitude regions contain abundant snow cover and glacial resources, and they are the source regions for many rivers.
- b) where  $SD_{sp}$  is the sub-pixel daily SD with a 500 m spatial resolution, *SD* is the daily SD with a 25 km spatial resolution, *SDYi* is the average number of SCDs for each MODIS

pixel in year *i*, and  $SDT_i$  is the sum of the total SCDs for each SD pixel in year i.

regions with more than 350 SCDs as permanent snow-covered regions.

- c) Considering the accuracy of the MODIS snow product (Wang et al., 2015), we classified
- 255

275

d) We used the M-K method to analyze the variation in the number of snow-covered days in the different seasons of winter (December–February next year), spring (March–May),

(2) Comments from Referees: 2. Some detailed problems in figures. a) Resolution of figures

summer (June-August), and fall (September-November) in the grid cells.

(dpi) should be enhanced, especially the maps. b) In figure titles, when it refers to "average annual", it is suggested to add time duration. Take Figure 3 for example, it is advised to be:
... annual average snow depth in China from 2001to 2014".

Author's Response: The resolution of each figure was enhanced by 400 dpi, hope can meet the Journal's requirement. And the time duration of each figure title was also added based on

265 your comments. Thank you so much.

Author's changes in manuscript:

- a) Resolution of figures was enhanced with 400 dpi.
- b) The time duration added for each figures.

(3) Comments from Referees: 3. There are some strange "missing" words or blanks

270 throughout the manuscript. a) Line 173, Line 177: "at a given significance level " b) Line 233: "( <0) "

Author's Response: The missing words or blanks throughout the manuscript were modified. Author's changes in manuscript:

a) At a given significance level  $\alpha$ , if  $|S| \ge S\alpha_{/2}$ , the trend of the series is significant; otherwise, it is insignificant.

b) At a given significance level  $\alpha$ , we looked up the critical  $Z\alpha_{/2}$  in the normal distribution table.

(4) Comments from Referees: 4. There are some leap years during the study period, but it seems that you assumed every year to be 365 days. Explanations are needed.

Author's Response: Among them, the 2004, 2008 and 2012 are leap years. The average SCA

refers the mean of 366 days.

Author's changes in manuscript: Fig. 2 summarizes the average annual SCA between 2001 and 2014. Leap years occurred in 2004, 2008 and 2012, so the average SCA refers to the

mean of 366 days for these years.

(5) Comments from Referees:5. As you speak highly of the M-K method in analyzing thevariation and trend of snow cover data, why you used Sen's median method "to test the accuracy of this result" (Line 230)? Do you have any explanations?

Author's Response: The M-K method can test the variation and trend of the snow cover, but can't examine the slope of the variation of the snow cover. We are sorry about the wrong statement of "to test the accuracy of this result" in the manuscript. The purpose of the Sen's median method used in the paper was to calculate the slope of the variation in the SCD.

- 290 median method used in the paper was to calculate the slope of the variation in the SCD. Author's changes in manuscript: The results of the M-K variation analysis showed that the annual number of SCDs in South China increased significantly. To further analyze the trend of the SCDs in China over the past 14 years, we calculated the slope of the variation in the annual SCDs using Sen's median method.
- (6) Comments from Referees: 6. The long time series of snow depth in China you used in WESTDC have been updated based on the following publications: a) Che, T., Dai, L.Y., Zheng, X.M., Li, X.F., Zhao, K., 2016. Estimation of snow depth from MWRI and AMSR-E data in forest regions of Northeast China. Remote Sensing of Environment 183, 334-349. b) Dai, L., Che, T., Ding, Y., 2015. Inter-Calibrating SMMR, SSM/I and SSMI/S Data to
- Improve the Consistency of Snow-Depth Products in China. Remote Sensing 7, 7212. c) Dai,
   L.Y., Che, T., Wang, J., Zhang, P., 2012. Snow depth and snow water equivalent estimation
   from AMSR-E data based on a priori snow characteristics in Xinjiang, China. Remote
   Sensing of Environment 127, 14-29.

Author's Response: Updated as your suggestion, thanks a lot.

305 Author's changes in manuscript:

New references updated are as follows:

Che, T., Dai, L.Y., Zheng, X.M., Li, X.F., Zhao, K.: Estimation of snow depth from MWRI and AMSR-E data in forest regions of Northeast China. Remote Sens. Environ, 183, 334-349, 2016.

Dai, L., Che, T., Ding, Y.: Inter-Calibrating SMMR, SSM/I and SSMI/S Data to Improve the Consistency of Snow-Depth Products in China. Remote Sens, 7, 7212-7230, 2015.
 Dai, L.Y., Che, T., Wang, J., Zhang, P.: Snow depth and snow water equivalent estimation

from AMSR-E data based on a priori snow characteristics in Xinjiang, China. Remote Sens. Environ, 127, 14-29, 2012.

- Hall, D. K., Riggs, G. A., Salomonson, V. V., Digirolamo, N. E., & Bayr, K. J.: MODIS snow-cover products. Remote Sens. Environ, 83(1): 181-194, 2002.
  Qian, Y. F., Zheng, Y. Q., Zhang, Y., Miao, M. Q.: Responses of China's summer monsoon climate to snow anomaly over the Tibetan Plateau. Int. J. of Climatol., 23, 593-613, 2003.
  Zhao, P., Zhou, Z. J., Liu, J. P.: Variability of the Tibetan spring snow and its associations
- with the hemispheric extropical circulation and East Asian summer monsoon rainfall: An observational investigation. J. Climate, 20, 3942-3955, 2007.
  Ke, C. Q., Li, X. C., Xie, H. J., Ma, D. H., Liu, X., Kou, C.: Variability in snow cover phenology in China from 1952 to 2010. Hydrol. Earth Syst. Sci., 20, 755-770, 2016.

**325 Spatio-temporal dynamics of snow cover based on multi-source remote sensing data in China**

Xiaodong Huang1, Jie Deng1, Xiaofang Ma1, Yunlong Wang1, Qisheng Feng1, Xiaohua Hao2, Tiangang Liang1

1Key Laboratory of Grassland Agro-Ecology System, College of Pastoral Agriculture Science and Technology, Lanzhou University, Lanzhou 730020, China;

2Chinese Academy of Sciences, Cold and Arid Regions Environmental and Engineering Research Institute, Lanzhou 730000, China

Correspondence to: Xiaodong Huang (huangxd@lzu.edu.cn)

335 Abstract. Through-By combining optical remote sensing snow cover products with passive microwave remote sensing snow depth (SD) data, we produced a MODIS cloudless binary snow cover product and a 500 m spatial resolution snow depth product. The temporal and spatial variations of snow cover from December 2000 to November 2014 in China were analyzed. The results indicated that, over the past 14 years, (1) the mean snow-covered area (SCA) in China was 11.3% annually and 27% in winter 340 season, with the mean SCA decreasing in summer and winter seasons, in increasing in spring and fall seasons, The average the summer and winter decreased, whereas the average the spring and fall increased and no much change annually; (2) the snow-covered days (SCDs) showed increasing in winter, spring, and fall, and annually, whereas decreasing in summer; (3) The the average SD decreased inthe winter, summer, and fall-decreased, while increased in the average spring and annually; 345 increased (4) The 
[revised manuscript text omitted]
 snow depthSD product: the long time series snow depthSD database of China was used to determine-identify cloud pixels, completely reclassified reclassify the residual cloud pixels to land or snow pixels, and produced the MODIS daily 450 cloudless binary snow cover images. Based on the downscaling algorithm for the AMSR-E snow water equivalent product by Mhawej et al. (2014), we conducted applied a downscaling algorithm toon the passive microwave snow depthSD product and built the 500-m spatial resolution snow depthSD data
  - in of China from for December 2000 to November 2014. The calculation equation is as follows:

$$\begin{cases} \text{if MODIS} = 0\\ SD_{sp} = 0\\ \text{else}\\ SDsp = \frac{SD \times SDY_i \times 2500}{SDT_i}, \end{cases}$$
(1)

where  $SD_{sp}$  is the sub-pixel daily snow depthSD with a 500 m spatial resolution,  $SD_i$  is the daily snow depthSD with a 25 km spatial resolution, SDYi is the average number of SCDs for each MODIS pixel in year *i*, and *SDTi* is the sum of the total SCDs for each SD pixel in year i.

**2.4 Analysis of the snow cover variation**

- 460 The Mann-Kendall (M-K) method is a nonparametric test method widely used in the analysis of long time series of data (Helsel and Hirsch, 1992). This method monitors the variation of monotonic nonlinear data. It has no requirement for the data distribution, and it can avoid the interference of a few anomalies (Mcbean and Motiee, 2008). This study used the M-K method to analyze the trend and significance level of the SCDs and SD in China at a-the pixel scale. For a series  $X_i = (X_1, X_2, ..., X_n)$ with n samples, the test process is as follows:
- 465

$$Z = \frac{S}{\sqrt{VAR(S)}}$$
(2)

where:

$$S = \sum_{i=1}^{n} \sum_{j=i+1}^{n} sgn(X_j - X_i)$$
(3)

$$\operatorname{sgn}(X_{j} - X_{i}) = \begin{cases} +1, if(X_{j} - X_{i}) > 0\\ 0, if(X_{j} - X_{i}) = 0\\ -1, if(X_{j} - X_{i}) < 0 \end{cases}$$
(4)

470
$$\operatorname{VAR}(S) = \frac{n(n-1)(2n+5) - \sum_{i=1}^{m} t_i(t_i-1)(2t_i+5)}{18}$$
 (5)

where n is the year count (n = 14), m is the number of nodes (repetitive data groups) in the series, and  $t_i$ is the node width (the number of repetitive data points in the ith repetitive data group).

When  $n \le 10$ , we directly used the statistic S for the two-sided trend test. S > 0 represents an increase, S = 0 represents no variation, and S < 0 represents a decrease. At a given significance level  $\underline{\alpha}$ , if  $|S| \ge S\alpha_{/2}$ , the series trend of the series is significant; otherwise, it is insignificant.

When n > 10, the statistic S approaches the standardized normal distribution. We used the test statistic Z for the two-sided trend test. Z > 0 represents an increase, Z = 0 represents no variation, and Z < 0represents a decrease. At a given significance level  $\underline{\alpha}$ , we looked up the critical  $Z_{\alpha/2}$  in the normal distribution table. If  $|Z| > Z\alpha_{/2}$ , the series trend is significant; if  $|Z| \le Z\alpha_{/2}$ , the trend is insignificant. Sen's median method was also used to analyze the slope of the variation of in the annual SCDs. This method calculates the slope median slope of n(n-1)/2 pairs of combinations in a series of length n. The

480

calculation equation is:

$$\beta = \operatorname{Median}\left(\frac{x_i - x_j}{i - j}\right), i > j \tag{6}$$

where  $\beta > 0$  represents an increase in the trend, and  $\beta < 0$  represents a decrease in the trend.

**485 3 Results**

**3.1 Snow-Covered Area**

Fig. 1-2 summarizes the annual average annual SCA between in 2001 and 2014. Leap years occurred in 2004, 2008 and 2012, so the average SCA refers to the mean of 366 days for these years. The results indicated that the average annual snow covered areaSCA in China in 2001–2014 constituted 11.3% of the whole entire study region. In the past 14 years, tThe annual average annual SCA-slightly varied slightly over the past 14 years, but-it did not show a significant increase or decrease significantly.

---

## Author Response (AR2)

**Comments to the Author:**

The reviewers are basically satisfied with the revised manuscript. However, I have a few suggestions before the manuscript can finally be accepted for publication by The Cryosphere:

(1) Comments from Referees: What are the dashed-lines for in Fig. 2 and 3? By reading the figure caption, the readers cannot know what are these dashed lines for. The authors need to add more text in the caption to explain. If these dashed lines are linear trends, the authors need to show the significant test value (p value) and correlation coefficients.

Author's Response: The dashed-lines are linear tends, the p value and correlation coefficients are added in Fig. 2 and 3. Thank you for the comments.

Author's changes in manuscript:

[Figure]

Figure 2: Average annual SCA in China between 2001 and 2014.

[Figure]

Figure 3: Histograms of the average SCA in each season in China from December 2000 to November

2014. (a), (b), (c), and (d) are the average SCA in winter, spring, summer, and fall, respectively.

(2) Comments from Referees: Figure 3: suggest after (a), (b), (c), and (d), add winter, spring, summer, and autumn, respectively.

Author's Response: Did as your suggestion. Thanks.

Author's changes in manuscript: Please see Fig. 3 attached.

(3) Comments from Referees: Figure 4: the scale is too coarse. Suggest to remove the scale for snow covered days more than 180 days and expand the remaining.

Non-public comments to the Author:

I have these detailed suggestions:

1). Fig. 4, remove any scales >180 days, use the following: <10; 11-60; 61-120; 121-180; >180.

Author's Response: Did as your suggestion. But the snow-covered days bigger than 360d is discussed separately in the manuscript, so we reclassified the snow-covered days as the following: <10; 11-60; 61-120; 121-180; 180-350; >350. Thanks.

Author's changes in manuscript:

[Figure]

Figure 4: Spatial distribution of the average annual number of snow-covered days during 2001-2014 in

China.